# Differential spatiotemporal development of Purkinje cell populations and cerebellum-dependent sensorimotor behaviors

Gerrit Cornelis Beekhof[1†], Catarina Osório[1†], Joshua J White[1],
Scott van Zoomeren[1], Hannah van der Stok[1], Bilian Xiong[1],
Ingo HMS Nettersheim[1], Willem Ashwin Mak[1], Marit Runge[1],
Francesca Romana Fiocchi[1], Henk-Jan Boele[1,2], Freek E Hoebeek[1,3],
Martijn Schonewille[1]*

[1]Department of Neuroscience, Erasmus MC, Rotterdam, Netherlands; [2]Princeton Neuroscience Institute, Princeton, United States; [3]Department for Developmental Origins of Disease, University Medical Center Utrecht Brain Center and Wilhelmina Children's Hospital, Utrecht, Netherlands

*For correspondence:
m.schonewille@erasmusmc.nl

†These authors contributed equally to this work

**Abstract** Distinct populations of Purkinje cells (PCs) with unique molecular and connectivity features are at the core of the modular organization of the cerebellum. Previously, we showed that firing activity of PCs differs between ZebrinII-positive and ZebrinII-negative cerebellar modules (Zhou et al., 2014; Wu et al., 2019). Here, we investigate the timing and extent of PC differentiation during development in mice. We found that several features of PCs, including activity levels, dendritic arborization, axonal shape and climbing fiber input, develop differentially between nodular and anterior PC populations. Although all PCs show a particularly rapid development in the second postnatal week, anterior PCs typically have a prolonged physiological and dendritic maturation. In line herewith, younger mice exhibit attenuated anterior-dependent eyeblink conditioning, but faster nodular-dependent compensatory eye movement adaptation. Our results indicate that specific cerebellar regions have unique developmental timelines which match with their related, specific forms of cerebellum-dependent behaviors.

## Introduction

The parasagittal organization of the cerebellum is fundamental to confer specificity to the coordination and adaptation of behavior. This organization is based on cerebellar modules, that is, anatomical and functional units (*Apps and Hawkes, 2009*; *Voogd, 1964*; *White and Sillitoe, 2013*), known to control specific tasks such as limb and finger movement (*Horn et al., 2010*; *Martin et al., 2000*), compensatory eye movements (*De Zeeuw and Yeo, 2005*; *Graham and Wylie, 2012*; *Voogd et al., 2012*; *Sugihara et al., 2004*), and associative motor learning (*Attwell et al., 2001*; *Hesslow and Ivarsson, 1994*; *Jirenhed et al., 2007*; *Mostofi et al., 2010*; *Raymond et al., 1996*). Purkinje cells (PCs) from different modules not only express different levels of molecular markers (*Apps and Hawkes, 2009*; *Cerminara et al., 2015*), such as ZebrinII (*Brochu et al., 1990*), but also have different physiological properties (*Xiao et al., 2014*; *Zhou et al., 2014*), project to discrete targets in cerebellar and vestibular nuclei (*Garwicz and Ekerot, 1994*; *Sugihara et al., 2009*), receive climbing fiber (CF) input from unique subnuclei of the inferior olive (*Sugihara and Shinoda, 2004*; *Sugihara and Shinoda, 2007a*; *Voogd and Ruigrok, 2004*), are linked to specific muscle groups (*Ruigrok, 2011*; *Ruigrok et al., 2008*) and are differentially predisposed to degeneration in

neurodegenerative mouse models (*Sarna and Hawkes, 2003*). Although the role of PCs in cerebellar circuitry and motor behavior has been explored extensively in the adult, the mechanisms underlying early circuitry formation and its impact in early motor function have not been systematically investigated. Understanding earlier circuitry formation is crucial to deciphering the relationship between functional zones and cerebellum-dependent behavior.

PCs are general orchestrators of cerebellar circuit development (*Fleming and Chiang, 2015*). For example, PCs contribute to the proliferation of granule cells through the release of Sonic Hedgehog (*Lewis et al., 2004*), as well as the parasagittal organization of afferents (*Arsénio Nunes et al., 1988*) and interneurons (*Sillitoe et al., 2008*). Concurrent to influencing this variety of developmental processes, PCs undergo their own migration, monolayer organization and growth of their large planar dendritic trees and axonal arbors. Recent evidence suggests that transient disruptions in PC development can have lasting effects and influence the development of other brain areas (*Badura et al., 2018*; *Wang et al., 2014*). However, there is still a great deal unknown about the normal developmental timeline of the cerebellar principal neuron, the Purkinje cell. The timing of the PC birth is related to its ultimate placement in the cerebellar cortex, with earlier born PCs settling generally more laterally than later born PCs (*Hashimoto and Mikoshiba, 2003*; *Namba et al., 2011*; *Sillitoe et al., 2009*). Additionally, a precise reorganization from embryonic clusters of PCs into the parasagittal stripes of the mature cerebellum is indicative of a very straightforward process from embryonic origins into mature modules (*Fujita et al., 2012*). Ultimately, the anatomical location of a PC has a large impact on its function within the circuit, correlates with its intrinsic properties and is developmentally determined.

Here, we sought to test the hypothesis that cellular and physiological differences in subpopulations of PCs in mice are established early in postnatal development and contribute to the formation of early cerebellar sensorimotor function. *In vivo* recordings revealed that by postnatal day (P) 12 it is possible to observe ZebrinII expression-related differences in firing rate of both simple (SSs) and complex spikes (CSs), and *in vitro* recordings in ZebrinII-positive (Z+) lobule X and ZebrinII-negative (Z–) lobule III confirmed that these differences are intrinsically driven as reported in the adult mouse and rat (*Xiao et al., 2014*; *Zhou et al., 2014*). Furthermore, we show that both PC populations also differ in their timeline of dendritic (from P18) and axonal (from P14) maturation. CF translocation appears to occur earlier in the nodular regions (by P7) further suggesting differences of developmental timelines between distinct cerebellar regions. Finally, we show that young animals (P21-P25), compared to adults, display more effective adaptation of compensatory eye movements, which is controlled by Z+ modules in the vestibulocerebellum (*Zhou et al., 2014*; *Sanchez et al., 2002*). In contrast, young animals show a reduction in their learning rate during eyeblink conditioning, which is linked to anterior Z– modules (*Hesslow and Ivarsson, 1994*; *Mostofi et al., 2010*), further supporting the emerging concept of differences in circuitry maturation in distinct cerebellar regions. Overall, this study shows for the first time that PC subpopulations' developmental timelines shape unique cerebellar circuitries that underlie different maturational profiles of specific cerebellar functions.

## Results

### Developing Purkinje cells operate at different rates depending on their cerebellar location

Following our previous work (*Zhou et al., 2014*), we investigated when the differences between Z– and Z+ PC activity emerge during development. We first performed extracellular recordings *in vivo* in PCs of *Slc1a6-EGFP* or C57BL/6J awake mice (*Figure 1A₁*). *Slc1a6-EGFP* mice express enhanced green fluorescent protein (EGFP) under the *Slc1a6* promotor, an expression pattern that correlates with high levels of ZebrinII (aldolase C) (*Gincel et al., 2007*; *Gong et al., 2003*). PCs were identified during the recording by the presence of SSs and CSs, while the consistent presence of a pause in SS following each CS (i.e. climbing fiber pause [CF pause]) confirmed that the recording was obtained from a single unit (*De Zeeuw et al., 2011*; *Figure 1A₂-Figure 1—figure supplement 1A₁*). Additionally, it was confirmed that CSs cause a pause in SS activity as the interval between CSs and the following SS was longer than the interval between two SSs in virtually all recorded cells (*Figure 1—figure supplement 2A*). PC recording locations were determined with biocytin or Evans blue, and

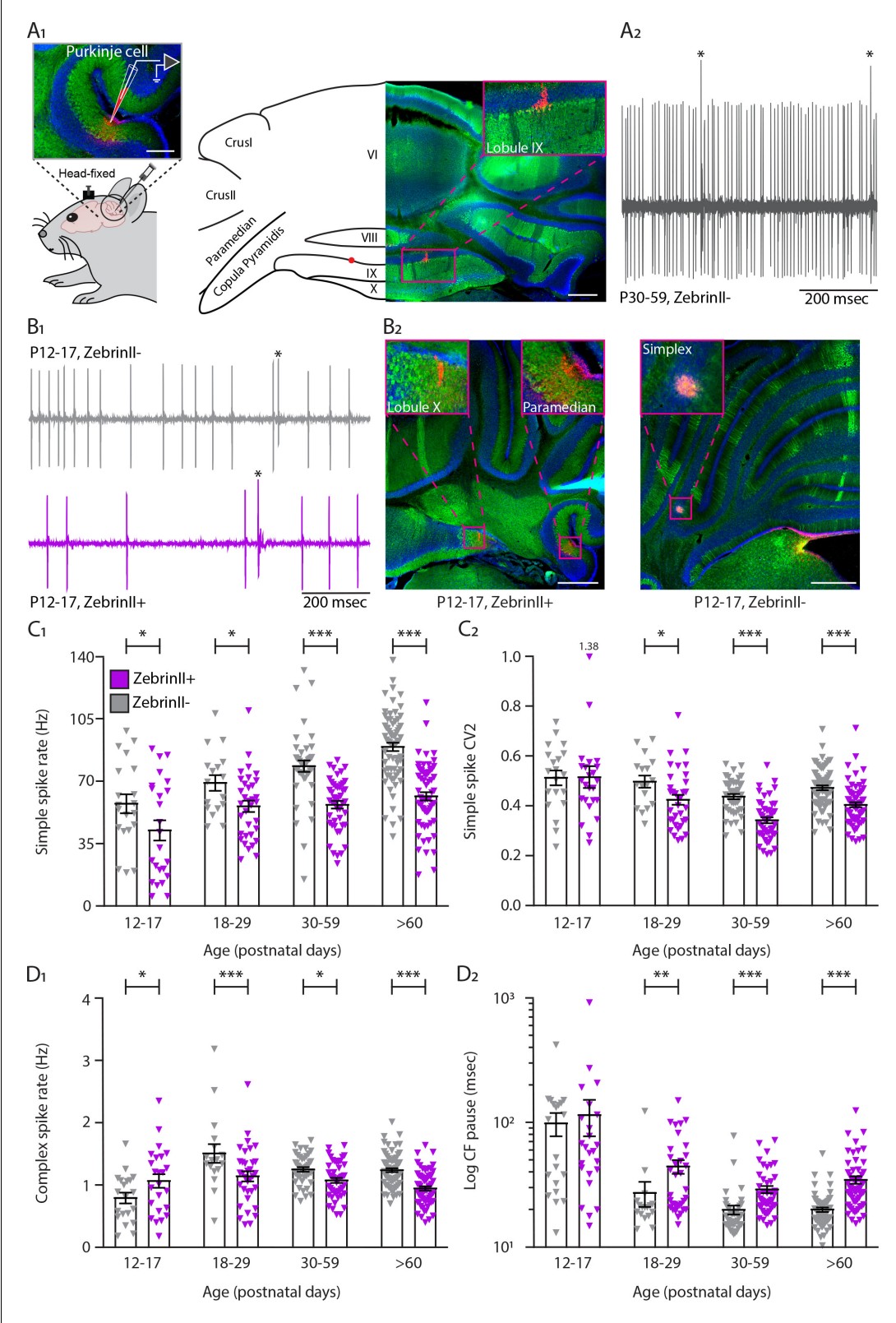

**Figure 1.** Simple and complex spike activity develops differently between ZebrinII-related Purkinje cell populations. (A₁) Schematic illustration of single unit cerebellar extracellular recordings in Purkinje cells of awake head-fixed mice (left). Schematic representation of a cerebellar coronal section and respective image showing lobules VI, VIII, IX and X, Crus I, Crus II, Paramedian, and Copula Pyramidis. Cerebellar coronal section with PCs labeled with Aldolase C (green). Inset showing recording site marked with biocytin (red) in lobule IX (right). (A₂) Example trace of *in vivo* P30-59 ZebrinII– Purkinje cell

*Figure 1 continued on next page*

*Figure 1 continued*

recording identified by its hallmark feature, the occurrence of complex spikes (asterisks) and simple spikes. (B$_1$) Example traces of extracellular Zebrin– and Zebrin+ Purkinje cell recordings at P12-17 with clearly distinguishable simple spikes and complex spikes (asterisks). (B$_2$) Photomicrographs of coronal sections with examples of ZebrinII+ (left) and ZebrinII– (right) P12-17 Purkinje cells in lobule X, Paramedian, and Simplex, respectively. Purkinje cells are labeled with Aldolase C (green). Insets showing recording sites marked with biocytin (red) in lobule X, Paramedian (left) and Simplex (right). (C$_1$) Purkinje cell simple spike firing rate, (C$_2$) coefficient of variation 2 (CV2) for simple spikes, (D$_1$) complex spike firing rate and (D$_2$) Climbing fiber pause recorded *in vivo* for ZebrinII– Purkinje cells (gray) and ZebrinII+ Purkinje cells (purple) in four age groups. Age groups: P12-17, P18-29, P30-59, and >P60. Error bars represent SEM., for values see *Supplementary file 1*. * denotes $p<0.05$, **$p<0.001$ and ***$p<0.0001$. Scale bars = (A$_1$) 100 μm (left), 500 μm (right), and (B$_2$) 500 μm.

The online version of this article includes the following source data and figure supplement(s) for figure 1:

**Source data 1.** Purkinje cell *in vivo* source data 1.

**Figure supplement 1.** *In vivo* extracellular recordings of simple and complex spike activity in different ZebrinII-related Purkinje cell populations during development.

**Figure supplement 2.** Climbing fiber pause is longer than inter-simple spike-interval in the majority of the recorded cells.

**Figure supplement 3.** *In vivo* extracellular recordings of simple spike activity distributed by lobules and cerebellar regions during development.

**Figure supplement 4.** Overview with color-coded simple spike and complex spike rate for all identified ZebrinII– and ZebrinII+ Purkinje cells.

**Figure supplement 5.** Overview with color-coded simple spike coefficient of variation 2 (CV2) and coefficient of variation (CV) for all identified ZebrinII– and ZebrinII+ cells.

their ZebrinII identity was determined histologically (*Figure 1B$_2$ -Figure 1—figure supplement 1A$_2$*). Immunohistochemical and electrophysiological data were divided in four different age groups: an early postnatal group of P12-P17, a juvenile group from P18 to P29, an adolescent group from P30 to P59, and an adult group from P60. We observed that SS firing rate is significantly higher in Z– PCs when compared with the Z+ PCs in all age groups, starting from early postnatal ages (P12-P17, Z–: 57.3 ± 5.3 Hz, Z+: 42.3 ± 5.7 Hz, $p=0.013$, see *Supplementary file 1* for additional age groups). Moreover, Z– PCs significantly increase their firing rate progressively from P12 until reaching their mature rate (P12-P17, Z– vs. >P60 Z–: $p<0.0001$) while Z+ PCs firing rate remained low and plateaued at P18, which is not significantly different from the rate at the mature stage (P18-P29 Z+ vs. Z+:>P60 Z+: $p=0.185$) (*Figure 1B$_1$,C$_1$*, *Figure 1—figure supplement 1A$_3$*, *Supplementary file 1* for additional age groups). Although there is no difference in SS regularity, measured as coefficient of variation 2 (CV2), in the young P12-P17 group (CV2, Z–: 0.51 ± 0.03, Z+: 0.51 ± 0.04, $p=0.94$), there is a significant difference between Z– and Z+ PC regularity in the older groups. From P18 onwards, Z– PCs are more irregular than Z+ PCs (>P60, Z–: 0.47 ± 0.01, Z+: 0.40 ± 0.01, $p<0.001$) (*Figure 1C$_2$*). Differences in the coefficient of variance (CV) between Z– and Z+ were only observed in the adult group (>P60, $p<0.0001$). Additionally, CV decreases from early postnatal onward in both groups (*Supplementary file 1*); as PCs become mature the variation in inter-spike-intervals (ISI) is reduced compared to young PCs (*Figure 1—figure supplement 1B$_1$*). The gradual shift to more regular activity was confirmed by a decreasing regularity index of SSs as the age increases for both groups, with only a significant higher level in Z+ PCs at P30-P59 ($p=0.011$, *Figure 1—figure supplement 1B$_2$*, *Supplementary file 1*).

Based on previous work (*Zhou et al., 2014*), we expected that the CS activity would be also lower in Z+ than in Z– PCs. CSs are driven by the activity of CFs that originate from the inferior olive. As expected, the CS activity of Z+ PCs is significantly reduced compared with Z– PC CSs in adulthood, a difference that becomes evident from as early as P18-P27 (Z–: 1.50 ± 0.15 Hz; Z+: 1.14 ± 0.08 Hz, $p<0.001$). In marked contrast, CS activity of Z+ PCs at P12-P17 is significantly higher when compared with Z– PC CSs (Z–: 0.79 ± 0.09 Hz; Z+: 1.06 ± 0.11 Hz, $p=0.011$) (*Figure 1D$_1$*). Interestingly, while in Z– PCs the CS rate is first significantly lower (P12-P17) and then higher (P18-P29) than in adult mice (>P60), the temporal profile of the CS rate in Z+ PCs shows less dramatic changes with only a significantly higher rate at P18-P29 (*Supplementary file 1*). Our previous data showed that at the adult stage, the duration of the pause in SS after a CS, the CF pause, was in line with the firing rate of SSs; a lower firing rate in Z+ PC would lead to a longer CF pause in these cells. In the developing cerebellum, several CFs innervate one single PC and synapse elimination occurs from P8 to P17 until a single selected CF innervates a single PC. The CF mono-innervation of a PC is established by the third postnatal week of cerebellar development (*Hashimoto and Kano, 2013*; *Watanabe and Kano, 2011*). Because the CF pause is longer in Z+ PCs than in Z– PCs in adult mice (*Zhou et al., 2014*), we hypothesized that longer CF pauses would be detected in Z+ PCs than in Z– PCs when CF

elimination is known to be completed at this mature stage. Our results confirmed our hypothesis. While there is no significant difference in CF pause in young cells P12-P17 (Z–: 81.1 ± 12.7 msec; Z+: 79.6 ± 13.8 msec, $p$=0.94), there is a significant increase in CF pause of Z+ PCs when compared with Z– PCs at P18, after synapse elimination, that is maintained into adulthood (>P60, Z–: 20.0 ± 0.8 msec; Z+: 34.5 ± 2.4 msec, $p$<0.0001) (*Figure 1D₂*). Additionally, CV2 of CSs is significantly decreased in Z+ PCs when compared with Z– PCs at P12-P17 (Z–: 0.91 ± 0.03; Z+: 0.85 ± 0.02, $p$=0.015), whereas no differences were observed in the intermediate groups. In the adult group (>P60), CV2 is significantly decreased in Z+ PCs when compared with Z– PCs (Z–: 0.84 ± 0.01; Z+: 0.81 ± 0.01, $p$=0.034) (*Figure 1—figure supplement 1C₁*). As for SS, the CS regularity index decreases with age in both groups, but no significant difference between Z– PCs and Z+ PCs was observed (all $p$>0.40; *Figure 1—figure supplement 1C₂*).

Although recordings were grouped based on ZebrinII expression, the observed developmental timeline could potentially also correlate with other factors. To examine this, we further subdivided recordings based on lobular identity. In adult mice, PCs in anterior, largely Z–, lobules I to III have higher SS firing rates than those in the nodular, largely Z+, lobules IX and X (*Figure 1—figure supplement 3A₁*, *Supplementary file 1*). For PCs recorded at P30-59 the pattern is similar, suggesting that in the latter stages of development, PC SS activity is also coupled to ZebrinII identity (*Figure 1—figure supplement 3A₂*, *Supplementary file 1*). For the P18-P29 and P12-P17 groups the sample sizes are insufficient to draw strong conclusions (*Figure 1—figure supplement 3A₃₋₄*). Hence, we employed a second approach, comparing the development of SS activity in two Z– regions, anterior lobules I to V vs. Z– hemisphere, and two Z+ regions, nodulus vs. flocculus regions. Although the number of samples prohibits strong conclusions, the developmental pattern of SS firing rate in the anterior and hemispheric region appears largely comparable, with significant increases toward adult levels, also after P12-P17 (*Figure 1—figure supplement 3 B1*). The development of activity is similar between the nodulus and flocculus as well, and best described by a stepwise increase from P12-P17 to P18-P29, after which the levels remain stable (*Figure 1—figure supplement 3 B2*). Thus, sample sizes hamper strong conclusions, as it is technically challenging to record PC activity *in vivo* without anesthesia, in for example P12-P17 mice. Finally, because some of the data set appear to be bimodal in this youngest group, we examined if at P12-P17 specifically there was any association between the PCs locations and their electrophysiological properties. Therefore, we mapped the location of P12-P17 PCs in the cerebellar cortex and their correspondent SS rate (*Figure 1—figure supplement 4A*), CS rate (*Figure 1—figure supplement 4B*), SS CV2 (*Figure 1—figure supplement 5A*), and SS CV values (*Figure 1—figure supplement 5B*). We find no evidence for bimodal distributions of data based on location of PCs.

Taken together, these results are in line with previous analyses in which we and others demonstrated the presence of differences across lobules in adult mice (*Xiao et al., 2014*; *Zhou et al., 2014*; *Chopra et al., 2020*). Our results suggest that differences in SS firing rate between Z– and Z+ PCs arise during development as early as P12 and the CS firing rate and CF pause settle at adult levels after the period of synapse elimination at P18. Overall, the firing rate of ZebrinII-identified PC populations differentiates from early postnatal ages, and reached a stable, adult level in Z+ PCs first.

## Developing Purkinje cells of distinct cerebellar regions have different intrinsic activity

PCs are intrinsically active in the absence of excitatory and inhibitory synaptic inputs (*De Zeeuw et al., 2011*; *Raman and Bean, 1999*; *Womack and Khodakhah, 2002*). In our previous work, we found that the difference in PC SS firing rate recorded *in vivo* was primarily the result of intrinsic activity of PCs in the adult mouse (*Zhou et al., 2014*). Next, we asked what is the contribution of intrinsic activity to the activity of developing PCs and when during development the differences in intrinsic activity arise among different populations of PCs. To answer these questions, we performed *in vitro* electrophysiological recordings on sagittal cerebellar sections of P3 to adult mice to measure the intrinsic properties of PCs throughout cerebellar development (*Figure 2—figure supplement 1A*). Because ZebrinII parasagittal patterning is only complete around P12-P15 (*Brochu et al., 1990*; *Lannoo et al., 1991*) and ZebrinII labeling is ambiguous before P12, we focus our *in vitro* studies on lobules III and X (*Brochu et al., 1990*; *Zhou et al., 2014*; *Sugihara and Quy, 2007b*; *Figure 2A₁*), which can be readily identified in the cerebellum at all ages, are oriented in such a way that it is possible to record from them in a single sagittal slice and have previously been demonstrated to be

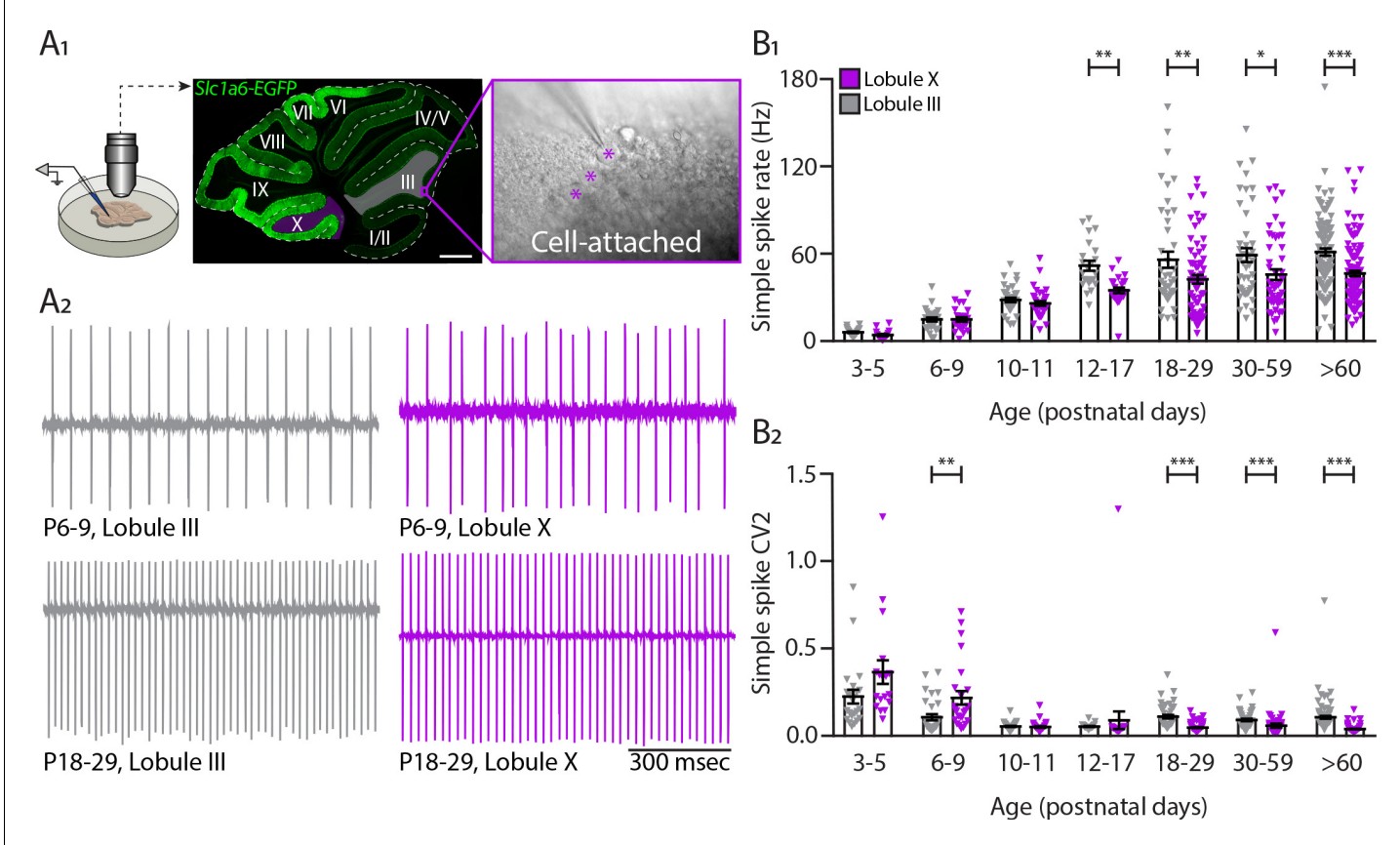

**Figure 2.** Intrinsic activity develops differently between lobule III and lobule X Purkinje cell populations. (A₁) Cell-attached recordings *in vitro* in *Slc1a6-EGFP* mice that express eGFP in Purkinje cells. Recordings of intrinsic Purkinje cell activity were made in lobules III and X. Purkinje cells were identified by the size of the soma (asterisk) clearly visible in the Purkinje cell layer. (A₂) Example traces of cell-attached recordings of lobule III and X Purkinje cells at P6-9 and P18-29. (B₁) Purkinje cell intrinsic simple spike rate and (B₂) coefficient of variation 2 (CV2) recorded *in vitro* for lobule III Purkinje cells (gray) and lobule X Purkinje cells (purple) in seven age groups. Age groups: P3-5, P6-9, P10-11, P12-17, P18-29, P30-59, and >P60. Error bars represent SEM, for values see *Supplementary file 1*. * denotes $p<0.05$, **$p<0.001$, and ***$p<0.0001$. Scale bar = (A₁) 500 µm.

The online version of this article includes the following source data and figure supplement(s) for figure 2:

**Source data 1.** Purkinje cell *in vitro* source data 1.
**Figure supplement 1.** *In vitro* cell-attached recordings of intrinsic activity in different Purkinje cell populations during development.
**Figure supplement 2.** Lobule III and ZebrinII– Purkinje cell simple spike rate results from a higher intrinsic and input driven component when compared with lobule X and ZebrinII+ Purkinje cells.

representative of the Z– and Z+ PC populations (*Zhou et al., 2014*; *Wu et al., 2019*). To remove the impact of synaptic inputs to PCs, we used blockers for NMDA, AMPA, GABAₐ and glycine receptors during the cell-attached recordings. For this experiment, we analyzed seven different age groups: P3-P5, P6-P9, P10-P11, P12-P17, P18-P29, P30-P59 and an adult group >P60. PC firing rate significantly increases from P3-P5 to P10-P11 (LIII $p=0.0002$; LX $p=0.0007$), but over these first days of development there were no differences in the firing rate between PCs of different cerebellar regions (*Figure 2A₂,B₁-Figure 2—figure supplement 2B*, *Supplementary file 1*). Starting from P12 to P17, similar to the *in vivo* data, we observed that the intrinsic activity of PCs located in lobule X is significantly different from those located in lobule III (LIII: 51.7 ± 3.4 Hz; LX: 34.8 ± 2.0 Hz, $p=0.008$, *Figure 2B₁*). After this key point during development, the differences in intrinsic firing rate in PCs from both cerebellar regions are maintained into adulthood (>P60, LIII: 61.2 ± 2.5 Hz; LX: 46.3 ± 1.9 Hz, $p<0.0001$). In contrast to the results *in vivo*, SS rate *in vitro* are both only significantly different from adult rates until P12-P17. However, the strongest increase in rate occurs later in lobule III PCs (P10-P11 to P12-P17:+23.5 Hz) than in those in lobule X (P6-P9 to P10-11:+11.1 Hz) (*Supplementary file 1*). In line with the *in vivo* results (*Figure 1B₂*), there is a significant increase in

irregularity in lobule III PCs from P18 when compared with lobule X PCs (CV2, LIII: 0.11 ± 0.01; LX: 0.05 ± 0.00, $p<0.0001$) and this is maintained into adulthood (>P60, CV2, LIII: 0.11 ± 0.01; LX: 0.04 ± 0.00, $p<0.0001$) (*Figure 2B$_2$*). These differences in CV2 are confirmed by lower CV and higher regularity index levels for lobule X PCs when compared with lobule III PCs from P18 to P29 onwards (P18-P29 CV, LIII: 0.14 ± 0.01; LX: 0.10 ± 0.01, $p<0.007$; P18-P29 regularity index, LIII: 0.016 ± 0.002; LX: 0.060 ± 0.004, $p<0.0001$) (*Figure 2—figure supplement 1C$_{1–2}$*). Young PCs at P6-P9 exhibit a significantly higher CV and higher regularity index in lobule III PCs when compared with lobule X PCs, suggesting the potential presence of subtler differences in early development (*Figure 2—figure supplement 1C$_{1–2}$*). All together, these data indicate that differences in SS firing rate at least in lobules III and X are indeed the result of differences of intrinsic activity of distinct PC subpopulations starting at P12-P17 and maintained into mature ages.

However, one cannot exclude the role of extrinsic input in PC SS rates. As observed in *Figure 2—figure supplement 2*, in both lobule III/Z– and lobule X/Z+ PCs, the firing rate *in vivo* was significantly higher than the firing rate *in vitro* (*Figure 2—figure supplement 2A$_1$*). This difference is evident from P18-P29 onwards. However, there was a higher input component in the lobule III/Z– when compared with lobule X/Z+ PCs (*Figure 2—figure supplement 2A$_2$*) contributing to the regional difference between the two subtypes of PCs.

## Dendrite complexity is more pronounced in Purkinje cells in the anterior lobule III

Neuronal morphology is a key determinant of the functional properties of neurons. Several studies have shown that different cell types show a causal relationship between firing patterns and neuronal morphology such as dendritic structure (*Mainen and Sejnowski, 1996*; *Vetter et al., 2001*), dendritic size (*Gollo et al., 2013*), and branching points (*Ferrante et al., 2013*). Hence, we hypothesized that differences in activity between different cerebellar regions could suggest differences in PC morphology. To investigate this possibility, we analyzed dendritic arborization complexity in different PC subpopulations of adult mice and during development. PCs from lobules I/II/III and IX/X were filled with biocytin and stained with Cy3-streptavidin. We then performed sholl analysis (*Ferreira et al., 2014*) to quantify PC dendritic arbor complexity, maximum length from the cell soma and cell area. Concomitantly with the physiology data, we observed that from P6-P11 to P12-17 there is a rapid increase in dendritic complexity and area (*Figure 3A*) across all PCs. However, unlike in the physiological data there is no significant difference in dendritic complexity (P12-P17, LI-III: 336.1 ± 31.7; LIX-X: 291.2 ± 17.1, $p=0.38$, *Figure 3B$_1$*), maximum length (P12-P17, LI-III: 138.3 ± 5.1 μm; LIX-X: 142.5 ± 3.9 μm, $p=0.74$, *Figure 3B$_2$*), and area (P12-P17, LI-III: 5419 ± 399 μm$^2$; LI-X: 6109 ± 462 μm$^2$, $p=0.46$, *Figure 3B$_3$*) between PCs in anterior (lobules I/II/III) and nodular (lobules IX/X) regions at P12-P17. While dendritic arborization in lobules IX-X marginally changed after this time point, dendritic complexity significantly increased in lobules I-III from the third postnatal week (from P18) into adulthood: dendritic complexity (P60, LI-III: 663.1 ± 23.8; LIX-X: 445.7 ± 22.2, $p<0.0001$, *Figure 3B$_1$*), maximum length (P60, LI-III: 206.6 ± 5.9 μm; LIX-X: 169.9 ± 4.0 μm, $p<0.0001$, *Figure 3B$_2$*), and area (P60, LI-III: 10890 ± 412 μm$^2$; LIX-X: 8520 ± 377 μm$^2$, $p<0.0001$, *Figure 3B$_3$*). A difference in size and complexity in dendritic arborization of PC subpopulations can therefore be first observed at P18 (see *Supplementary file 1*).

Because PC morphology can vary depending on their location in the same lobule during development (*Nedelescu and Abdelhack, 2013*; *Nedelescu et al., 2018*; *Sudarov and Joyner, 2007*), we investigated the location of each PC analyzed per age in the apex, base and sulcus of each lobule (*Figure 3—figure supplement 1A*). There was not systematic bias in our sampling from apex, base or sulcus (*Figure 3—figure supplement 1B$_1$*) at different developmental ages and between anterior and nodular lobules (*Figure 3—figure supplement 1B$_2$*). Additionally, most of the cells collected for this analysis were located in lobule III and X (*Figure 3—figure supplement 1C$_{1–2}$*) at different developmental ages.

Our data show that indeed the rapid period of maturation for PCs occurs between P9 to P12. The slow period of dendritic expansion however differs between PC populations. PCs located predominantly in lobule III grow larger and have more complex dendritic trees compared to PCs located predominantly in lobule X. These data also suggest that, in line with their firing rates, PCs in lobule X reach their dendritic maturity earlier than PCs from lobule III.

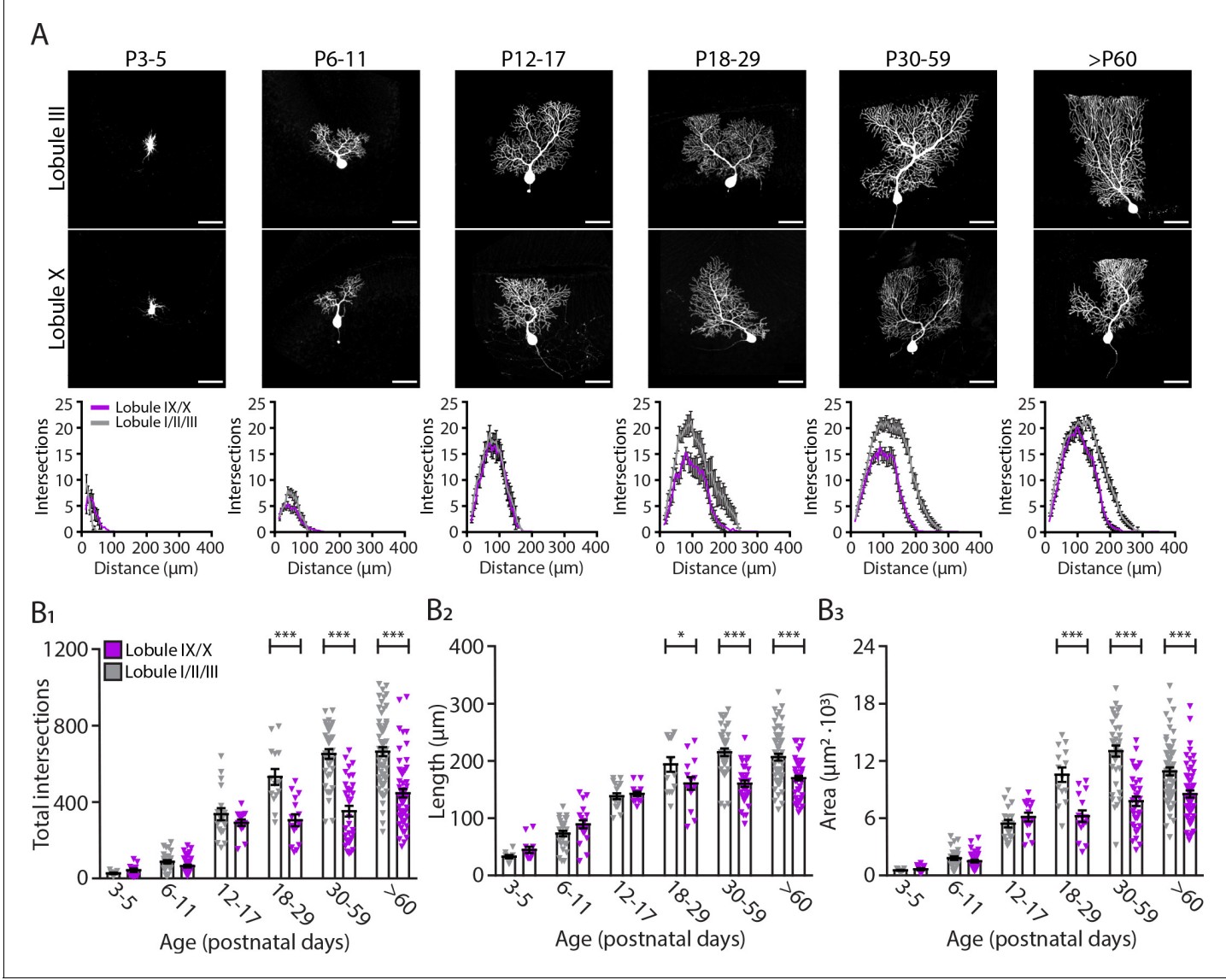

**Figure 3.** Purkinje cells located in the lobule III display a larger and elaborated dendritic tree. (**A**) Photomicrographs of Purkinje cells filled with biocytin in lobule III and X (top panel); and sholl analysis of Purkinje cells located in lobules I/II/III (gray) or IX/X (purple) in six age groups (lower panel). (**B₁**) Total number of intersections, (**B₂**) longest dendrite length and (**B₃**) area analysis for lobules I/II/III Purkinje cells (gray) and lobules IX/X Purkinje cells (purple) in six age groups. Age groups: P3-5, P6-11, P12-17, P18-29, P30-59, and >P60. Error bars represent SEM, for values see *Supplementary file 1*. * denotes $p < 0.05$, **$p < 0.001$, and ***$p < 0.0001$. Scale bar = (**A**) 50 µm.

The online version of this article includes the following source data and figure supplement(s) for figure 3:

**Source data 1.** Purkinje cell dendritic morphological source data 1.

**Figure supplement 1.** Purkinje cell location in the cerebellum within each lobule.

**Figure supplement 1—source data 1.** Apex, Base, Sulcus source data 1.

## Translocation of climbing fibers occurs earlier in the nodular cerebellum

Both our physiological and morphological data suggest that PCs from the anterior lobule III, and Z– PCs recorded *in vivo*, reach mature levels later when compared to PCs from lobule X, or the Z+ population. Because CF and PC development are intertwined (*Watanabe and Kano, 2011*), we tested for potential differences between modules in CF development. To characterize and compare CF development in lobules I-III and IX-X, we stained PCs with calbindin and CFs with VGluT2, which in the molecular layer labels CF terminals, at P7, P14, P21, P35, and P60 (*Figure 4A*). Remarkably, we found that at P7 there was a significant increase in VGluT2 puncta in lobules IX-X (P7, LI-III: 0.62 ±

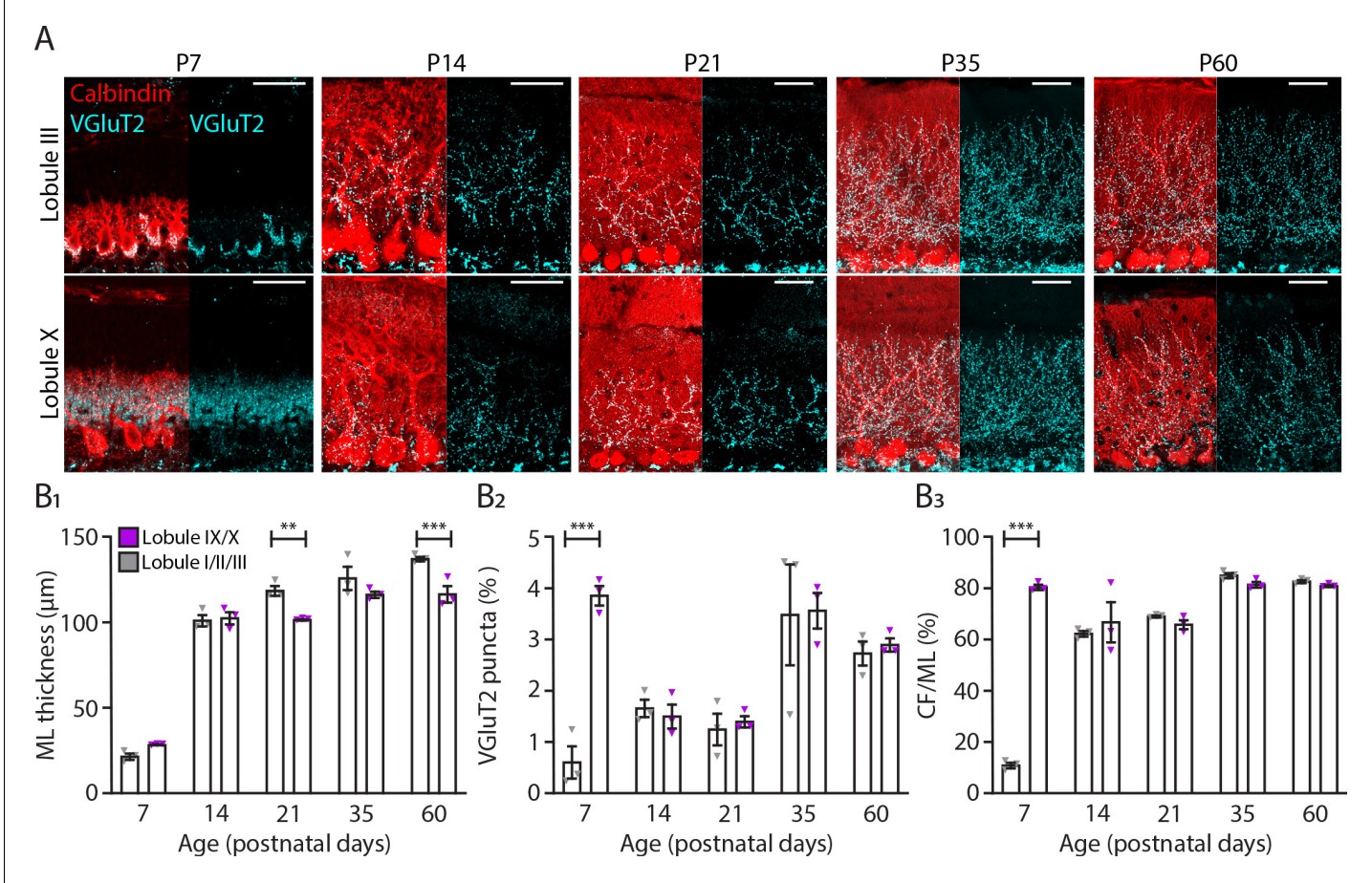

**Figure 4.** Translocation of climbing fibers occurs earlier in the nodular regions of the cerebellum. (A) Photomicrographs of lobule III and lobule X regions depicting climbing fiber (VGluT2 staining, cyan) and Purkinje cell (calbindin, red) development in five age groups. (B₁) Molecular layer (ML) thickness, (B₂) VGluT2 puncta per area of region of interest and (B₃) climbing fiber (CF) extension per ML thickness quantification for lobules I/II/III (gray) and lobules IX/X (purple) in five age groups. Age groups: P7, P14, P21, P35, and P60. Error bars represent SEM, for values see *Supplementary file 1*. * denotes $p<0.05$, **$p<0.001$, and ***$p<0.0001$. Scale bar = (A) 40 µm.

The online version of this article includes the following source data for figure 4:

**Source data 1.** VGluT2 puncta source data 1.
**Source data 2.** Molecular layer thickness and climbing fiber height source data 2.

0.31%; LIX-X: 3.87 ± 0.19%, $p<0.0001$, ***Figure 4B₂***) when compared with lobules I-III. As a result, at P7 the percentage of CF extension into the molecular layer (ML) was significantly increased in lobules IX-X (P7, LI-III: 11.1 ± 1.1%; LIX-X: 80.5 ± 1.0%, $p<0.0001$, ***Figure 4B₃***). These results suggest that CF translocation starts earlier in nodular lobules (***Figure 4A***). In anterior lobules, most of the VGluT2 puncta at P7 were still localized around the PC somata while in nodular lobules the CF terminals were predominantly targeting the younger dendritic arbors. From P7 to P14, there was a dramatic increase in ML thickness in both regions of the cerebellum (***Figure 4B₁***) but the significant difference in VGluT2 puncta and CF terminals disappeared (VGluT2 puncta, P7: $p<0.0001$, P14: $p=0.78$; CF terminals, P7: $p<0.0001$, P14: $p=0.24$, ***Figure 4B₂ and B₃***). From P21 into adulthood, there was a significant increase in ML thickness in lobules I-III when compared with the nodular lobules (P60, LI-III: 137.5 ± 1.3 µm; Lob IX-X: 116.9 ± 4.9 µm, $p<0.001$, ***Figure 4B₁***). With the exception of an increase in VGluT2 puncta and CF terminals from P21 to P35 for both regions studied, no other differences were observed (***Figure 4B₂ and B₃***; ***Supplementary file 1***). These results indicate that CF translocation into the PC dendrite starts earlier in nodular lobules when compared with

anterior lobules I, II, and III pointing to another difference in the maturation of the cerebellar circuitries in different cerebellar regions.

## Purkinje cell axonal complexity increases during the second postnatal week

Our data demonstrate that intrinsic physiology of PCs as well as their input structure, the dendritic tree, develop differentially between anterior/Z− and nodular/Z+ populations as well as exhibiting a stark growth period beginning in the second half of the second postnatal week. We next asked whether the PC output structure, the axonal arbor, matches this developmental timeline. The PC axon is a large structure that targets specific cerebellar nuclei based on their location and ZebrinII identity (*Sugihara et al., 2009*). PC axons are present in the cerebellar nuclei as early as embryonic day (E) 15.5 in mice (*Sillitoe et al., 2009*) and E18 in rats (*Eisenman et al., 1991*). Targeting to the correct cerebellar subnuclei is also already in place at very early points in cerebellar development (*Sillitoe et al., 2009*). In culture, PC axons exhibit a multi-step developmental process (*de Luca et al., 2009*). However, the developmental process of PC axons *in vivo* is not known. Taking advantage of the *Pcp2-cre^ERT2;Ai14* mouse model, we sparsely labeled PCs with red fluorescent protein (RFP) and searched for isolated axonal arbors within the cerebellar nuclei (*Figure 5A*). Labeling was optimal between P10 and P21, although it was sometimes possible to identify well-isolated axons at P7 as well. We therefore analyzed axon arbor morphology as a total group at P7, P10, P14, and P21, while comparisons between Z− and Z+ subnuclei were restricted to P10, P14 and P21 (*Figure 5B*). Labeled PCs were randomly distributed throughout the cerebellar cortex (*Figure 5—figure supplement 1*). We performed three-dimensional sholl analysis on single axon arbors with the first branching point as the center. This analysis revealed significant differences across age of PC axon arbor maximum length and complexity, as both parameters increase from P7 to P14 and then decrease at P21 (number of intersections, P7: $84.2 \pm 18.2$; P10: $254.4 \pm 40.5$; P14: $277.4 \pm 17.5$; P21: $296.4 \pm 37.9$, p=0.013; length, P7: $124.2 \pm 17.8$ µm; P10: $212.7 \pm 24.4$ µm; P14: $216.4 \pm 14.3$ µm$^2$; P21: $225.2 \pm 16.1$ µm$^2$, p=0.060, *Figure 5C,E$_1$, E$_2$*). Area taken up by the axon arbors, however, continued to increase with age (area, P7: $1.52 \pm 0.20$ µm$^2 \cdot 10^3$ P10: $1.47 \pm 0.25$ µm$^2 \cdot 10^3$; P14: $2.26 \pm 0.19$ µm$^2 \cdot 10^3$; P21: $3.07 \pm 0.29$ µm$^2 \cdot 10^3$, p<0.0001; *Figure 5E$_3$*). The primary source of the peak at P14 is driven by Z− axon arbors (*Figure 5D$_2$*). Sholl analysis revealed significant differences between Z− and Z+ axon arbors at P10, P14 and P21, with Z− axon arbors exhibiting a denser branching complexity than Z+ (P10: p=0.0457; P14: p=0.0017; P21: p=0.0388). Significant differences were found between Z− and Z+ groups for length and area but not number of intersections (intersections, p=0.252; length, p=0.0197; area, p=0.0257; *Figure 5F$_{1-3}$*).

Taken together, our novel approach to analyze axon terminal development in cerebellar nuclei indicates that PC axons reach near mature shapes already early in development, with a peak in size around P14 and subtle differences between Z− and Z+ axons.

## Differential developmental timelines for the emergence of cerebellar-specific behaviors

Our data indicate that differentiation of cerebellar subpopulations can start early in development and suggest that different cerebellar regions have distinct developmental trajectories. To address if differences in cerebellar maturation have behavioral relevance, we aimed to study the performance of mice in cerebellum-dependent behavioral tasks that are directly linked to the PC subpopulations. The different developmental timeline of PC SS Activity was comparable between Z+ PCs in the flocculus and nodulus, as well as between Z− PCs in the hemispheres and anterior cerebellum (*Figure 1—figure supplement 3B*). Based on these observations, we performed two cerebellum-dependent learning tests, vestibulo-ocular reflex (VOR) adaptation and eyeblink conditioning (EBC), linked to the Z+ flocculus and the Z− anterior and hemispheric cerebellar regions, respectively. To facilitate the comparison, we determined the learning curves in juvenile mice directly after weaning (starting from P21) and for reference compared these curves to those of adult (10–11 weeks old) mice. The VOR ensures the stabilization of images on the retina via compensatory eye movement every time the vestibular system is activated (head movement). VOR adaptation is the adjustment of compensatory eye movements based on a mismatch between vestibular and visual input, moving in the same or opposite direction (i.e. in or out of phase) (*Ito, 1982*; *Nagao, 1989*; *Schonewille et al.,*

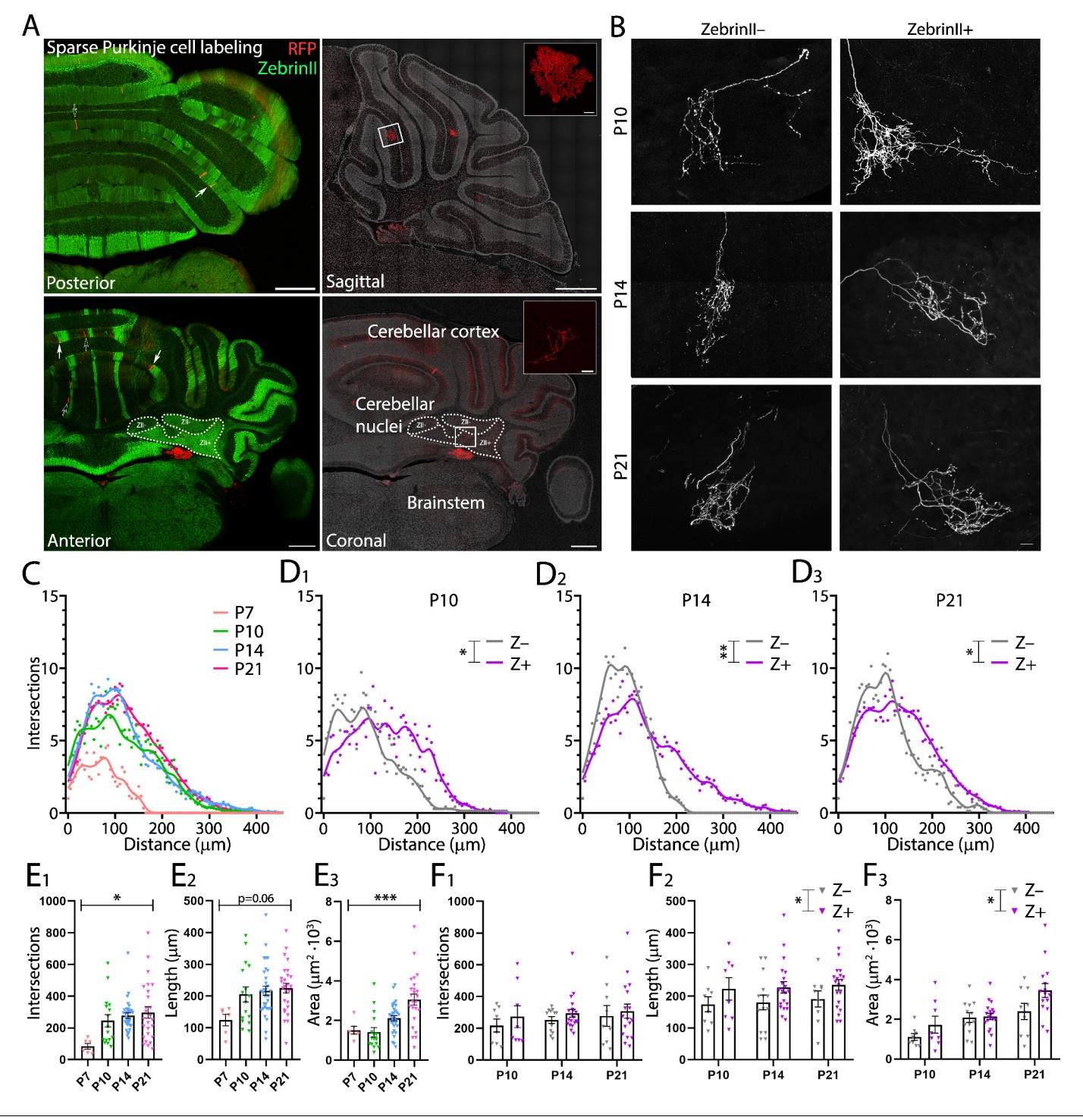

**Figure 5.** ZebrinII+ Purkinje cell axons are larger and less dense at all early postnatal ages. (**A**) Photomicrographs of cerebellar sections demonstrating sparse labeling of Purkinje cells. Left panels are stained with antibodies against ZebrinII as well as red fluorescent protein (RFP) to label Purkinje cells expressing RFP. Right panels are stained with DAPI and RFP to show examples of Purkinje cells and Purkinje cell axon arbors (top inset and bottom inset, respectively) in sagittal and coronal cerebellar sections (top and bottom right panels, respectively). (**B**) Photomicrographs of example ZebrinII– and ZebrinII+ axon arbors from P10, P14, and P21 mice. (**C**) Sholl analysis of all axon arbors at P7, P10, P14, and P21, average crosses are indicated with points and fitted with a smoothed line. (**D₁₋₃**) Sholl analyses of ZebrinII– and ZebrinII+ groups of axons at P10, P14, and P21, respectively. (**E₁**) Total number of intersections, (**E₂**) longest dendrite length and (**E₃**) axon arbor area analysis for all axons at different ages. (**F₁**) Total number of intersections, (**F₂**) longest dendrite length and (**F₃**) axon arbor area analysis for ZebrinII– (gray) and ZebrinII+ axon arbors (purple) at P10, P14, and P21. Error bars

*Figure 5 continued on next page*

*Figure 5 continued*

represent SEM., for values see ***Supplementary file 1***. * denotes *p<0.05*, **p<0.001*, and ***p<0.0001*. Scale bars = (**A**) 500 µm in large panels; 20 µm in top inset, 50 µm in bottom inset. (**B**) 20 µm.

The online version of this article includes the following source data and figure supplement(s) for figure 5:

**Source data 1.** Purkinje cell axonal morphological source data 1.

**Figure supplement 1.** Random distribution of labeled Purkinje cells for axon analysis.

*2010*; ***Wulff et al., 2009***). Moreover, compensatory eye movements are controlled by the flocculus, together with the nodulus forming the vestibulocerebellum, which is predominantly Z+ (***Sugihara and Shinoda, 2004***; ***Hawkes and Herrup, 1995***; ***Sillitoe and Hawkes, 2002***; ***Figure 6A_1***). In the EBC paradigm, a neutral sensory stimulus alone leads to a well-timed eyeblink after repeated pairing with a noxious stimulus which induces a reflexive eyelid closure (usually a mild airpuff) (***Boele et al., 2010***; ***McCormick and Thompson, 1984***; ***Thompson and Steinmetz, 2009***). The cerebellar eyeblink regions putatively reside in the anterior cerebellum, in particular at the border of lobule IV-V to VI, and from the hemisphere lobule IV-V to simplex, a region that is largely ZebrinII-negative (***Sugihara and Shinoda, 2004***; ***Hawkes and Herrup, 1995***; ***Sillitoe and Hawkes, 2002***; ***Heiney et al., 2014***; ***Figure 6B_1***).

We first tested if juvenile and adult animals differed in their basal optokinetic reflex (OKR; eye movements driven sole by visual input), vestibular-ocular reflex (VOR; eye movements driven sole by vestibular input in the dark) and visually enhanced VOR (VVOR, a combination of OKR and VOR in the light) (***Figure 6—figure supplement 1***). We determined the gain (the ratio of eye movement to stimulus amplitude) and phase (timing of the response relative to input) of basal compensatory eye movements at different oscillation frequencies (0.1–1 Hz). Neither the OKR gain or phase differ significantly between age groups (OKR gain, young: $0.60 \pm 0.09$; adult: $0.59 \pm 0.10$, p=0.84; and OKR phase, young: $-15.5 \pm 6.7$; adult: $-15.4 \pm 5.0$, *p=0.96*, ***Figure 6—figure supplement 1A_1,B_1***). In contrast, the VOR gain is significantly increased (VOR gain, young: $0.42 \pm 0.09$; adult: $0.61 \pm 0.09$, *p<0.0001*, ***Figure 6—figure supplement 1A_2***) and the VOR phase decreased (VOR phase, young: $29.4 \pm 5.8$; adult: $21.4 \pm 4.5$, *p=0.0024*, ***Figure 6—figure supplement 1B_2***) in adult mice when compared with juveniles. Moreover, while there are no differences in the VVOR phase (VVOR phase, young: $1.33 \pm 0.46$; adult: $1.35 \pm 0.13$, *p=0.97*, ***Figure 6—figure supplement 1B_3***), there is a significant increase in the VVOR gain of juvenile animals compared with adults (VVOR gain, young: $0.89 \pm 0.01$; adult: $0.96 \pm 0.01$, *p=0.009*, ***Figure 6—figure supplement 1A_3***).

Subsequently, we tested juvenile and adult mice on a phase-reversal VOR adaptation protocol. This test aimed to evaluate the ability of the animals to reverse the direction of their VOR, from the compensatory, normal direction, opposite to the head rotation, to moving in the same direction as the head (visual and vestibular stimulation in phase). Surprisingly, we observed that the training resulted in juvenile animals reversing their VOR phase, probed in the dark with only vestibular stimulation, faster with significantly lower gain values on the first 3 days and higher phase values on all days (VOR gain, days 1–3, all *p<0.01* phase reversal, all *p<0.0001*, ***Figure 6A_2*** and ***Figure 6—figure supplement 2A–B***). Multiple factors can underlie such a robust difference in VOR adaptation, including differences in (a) starting point, (b) gaze parameters, (c) path and speed of adaptation and (d) consolidation. With respect to the starting point (a), both the baseline data and pre-training values of the VOR phase reversal indicate that the juvenile mice start the training with a lower gain and higher phase value, suggesting that a difference in starting point contributed to the overall difference in the curve. It should be noted though, that on day 2 adult mice start with gain values similar to those of juvenile mice on day 1, and still display a slower learning curve from that point onwards. To evaluate the potential role of gaze (b) (***Shin et al., 2014***), we analyzed the eye movements during the training sessions when visual and vestibular input are combined (***Figure 6—figure supplement 2D–E***). Gain and phase values of juvenile are significantly closer to the optimal values based on the visual input, but the differences with adult mice are small relative to the differences observed in the VOR adaptation curves (***Shin et al., 2014***). Presenting compensatory eye movement data in separate gain and phase plots impedes the possibility to examine the trajectory, or 'tactics', employed by the mice to complete the task. To visualize the path and speed of adaptation (c), we re-plotted gain and phase in a polar plot (***Figure 6—figure supplement 3A–B***, respectively). Overall, eye movements of

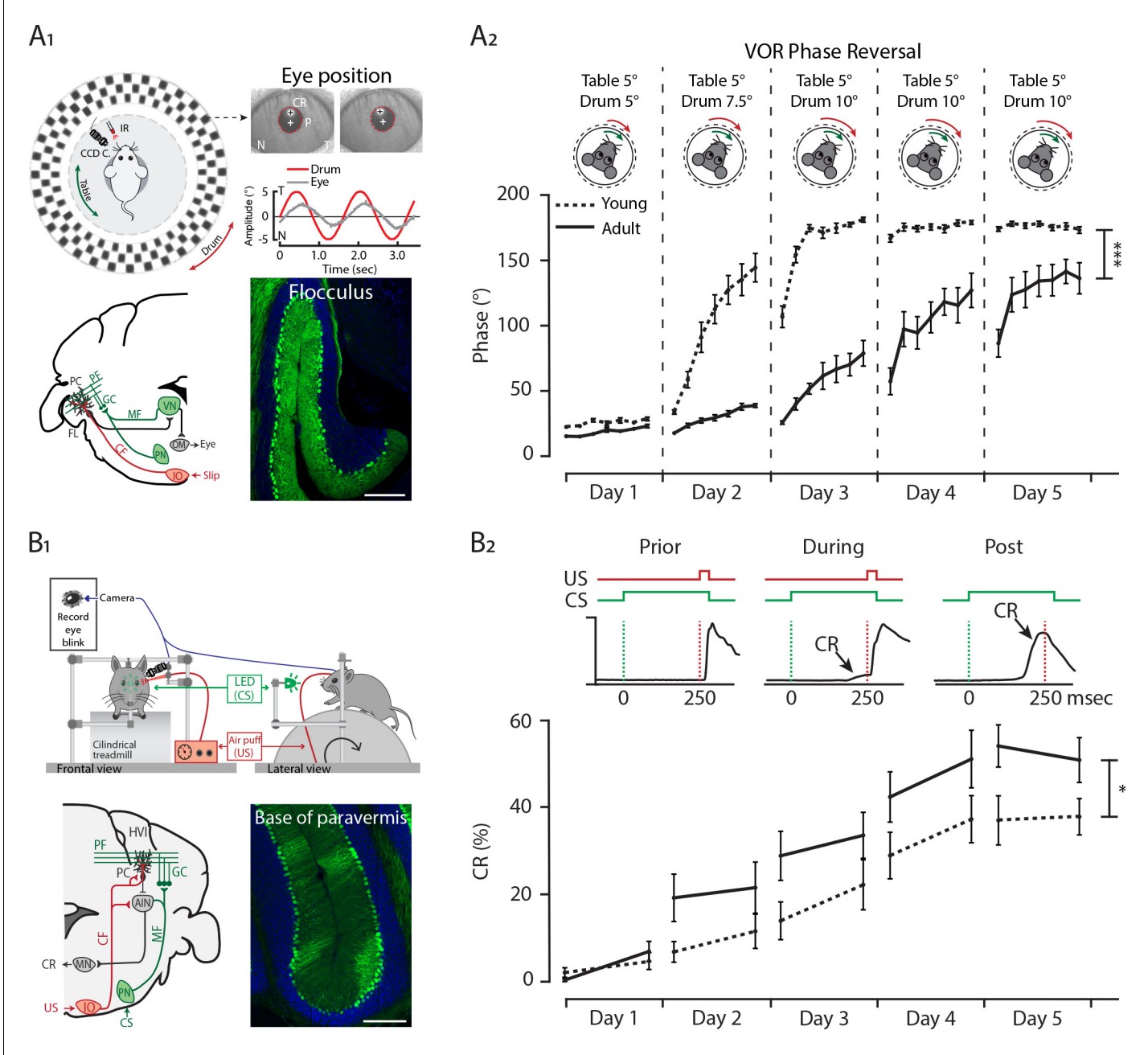

**Figure 6.** Differential development of cerebellar regions impacts the maturation of cerebellar-specific behaviors. (A₁) Schematic illustration of eye movement recording setup. Mice are head-fixed in the center of a turntable (green arrow) for vestibular stimulation and surrounded by a random dotted pattern *drum* (red arrow) for visual stimulation. A CCD camera was used for infrared (IR) video-tracking of the left eye (top left). Examples of nasal (N) and temporal (T) eye positions. Red circles = pupil fit; black cross = corneal reflection (CR); white cross = pupil (P) center. Example trace of eye position (gray) with drum position (red), during stimulation at an amplitude of 5° and frequency of 0.6 Hz (top right). Cerebellar circuitry controlling compensatory eye movements and their adaptation. Purkinje cells in the flocculus (FL) receive vestibular and visual input via the mossy fiber (MF) - parallel fiber (PF) system (green) and climbing fiber which influence eye movements via the vestibular nuclei (VN) and the oculomotor (OM) neurons. PN, pontine nuclei; GC, granule cell (bottom left). Photomicrograph of the flocculus a ZebrinII+ Purkinje cell predominant region; Purkinje cells labeled with Aldolase C (green) (bottom right). (A₂) Results of 5 days of vestibule-ocular reflex (VOR) phase reversal training, probed by recording VOR (in the dark before, between and after sessions) with mice kept in the dark in between experimental sessions in young (P21-25, dotted line) and adult (P70-90, full line) mice. (B₁) Schematic illustration of the eyeblink conditioning setup. Head-fixed mice on a freely moving treadmill, are presented a green LED light (conditioned stimulus, CS) followed several hundred milliseconds later by a weak air-puff on the eye (unconditioned stimulus, US). Eyelid movements were recorded with a camera (top). Cerebellar circuitry controlling eyeblink conditioning. Purkinje cells in the paravermal region around the primary fissure receive inputs carrying sensory information from for example the pontine nuclei (PN) through the mossy fiber-parallel fiber (MF-PF)

*Figure 6 continued on next page*

*Figure 6 continued*

pathway and the error signal from the inferior olive (IO) through the climbing fiber (CF). These Purkinje cells in turn influence eyelid muscles via the anterior interposed nucleus (AIN) and motor nuclei (MN) (bottom left). Photomicrograph of the base of paravermis a ZebrinII– Purkinje cell predominant region; Purkinje cells labeled with Aldolase C (green) (bottom right). (B$_2$) As a result of repeated conditioned stimulus (CS)-unconditioned stimulus (US) pairings, mice will eventually learn to close their eye in response to the conditioned stimulus (CS), which is called the conditioned response (CR) (top). Percentage of conditioned response (CR%) in young (dotted line) and adult (full line) mice during 5 days of training (bottom). Error bars represent SEM., for values see **Supplementary file 1**. * denotes $p<0.05$, **$p<0.001$, and ***$p<0.0001$. Scale bars = (A) 200 µm.

The online version of this article includes the following source data and figure supplement(s) for figure 6:

**Source data 1.** Eyeblink source data 1.
**Source data 2.** Eye movement source data 2.
**Figure supplement 1.** Compensatory eye movements in young and adult mice.
**Figure supplement 2.** Vestibulo-ocular reflex phase reversal gain, overnight consolidation, and gaze during training differ between juvenile and adult mice.
**Figure supplement 3.** Polar plot for vestibulo-ocular reflex phase reversal gain.
**Figure supplement 4.** Eyeblink conditioning in young and adult mice.

juvenile mice not only move faster from the pre-training position in the right top quadrant to the target gain and phase value left on the x-axis, but also follow a shorter path with lower gain values in the period that the phase increases, potentially indicating a different tactic for adaptation. Finally, we analyzed the consolidation of the adaptation (d) overnight, when animals were kept in the dark for 23 hr. We focused on consolidation of gain change from day 1 to day 2 and phase from day 2 to 3 and 3 to 4, to assure that there are sufficiently large adaptive changes in the comparisons. Consolidation is significantly larger in juvenile mice than in adult mice, for gain decrease ($p=0.0005$) and for phase increase ($p=0.0006$ and $p=0.0069$, respectively, *Figure 6—figure supplement 2C*). Taken together, these data indicate that the difference in VOR adaptation between juvenile and adult mice depends on multiple factors. While correcting for starting points would remove some of the difference, particularly the difference in consolidation, linked to cerebellar functioning (*Wulff et al., 2009*), is very robust.

Next, mice were trained to associate a 250 msec LED light (conditioned stimulus) and a 30 msec air puff delivered to the mouse cornea (unconditioned stimulus) co-terminating with the conditioned stimulus, which triggers an unconditioned response (UR), an eyeblink reflex. After training, the conditioned stimulus alone evokes perfectly timed conditioned responses (CR, preventative eyelid closure) (*Figure 6B$_1$*). Juvenile animals exhibit a significantly lower CR percentage after the first day of learning (juvenile: 20.4 ± 4.4%; adult: 30.9 ± 6.0%, p=0.027, *Figure 6B$_2$*) when compared with the adult animals. Moreover, the amplitude of the eyelid closure in response to the conditioned stimulus (fraction eyelid closure) is reduced in juvenile animals (*Figure 6—figure supplement 4A$_2$,C$_1$,C$_2$,C$_3$,C$_4$*). The difference in CR percentage was not due to reduced eyelid closure abilities in juvenile mice because UR timing and onset is similar between the two groups (UR onset, young: 5.61 ± 0.15; adult: 6.41 ± 0.10, p=0.41; UR peak time, young: 67.2 ± 1.0; adult: 69.0 ± 1.0, p=0.55, *Figure 6—figure supplement 4A$_1$,B$_1$,B$_2$*).

Taken together, our data indicate that juvenile animals had an attenuated rate of EBC acquisition while the VOR adaptation was heightened in juvenile animals when compared with adult animals. This suggests different developmental timelines in distinct cerebellar regions underlie the emergency of cerebellar specific behaviors.

## Discussion

While the variety of sensorimotor and cognitive functions controlled by the cerebellum and its intricate architecture have been widely studied in adult animals, the development of its regional differences in physiology, morphology and connectivity is more enigmatic. Identification of developmental milestones in different cerebellar regions is essential for deciphering the emergence of functional cerebellar circuits. In this study, we uncover that different subpopulations of PCs acquire unique morphological and physiological features during distinct developmental timelines, such that PCs of the posterior cerebellum, particularly lobule X, reach their adult stage prior to PCs of the anterior, in

e.g. lobule III, cerebellum (*Figure 7*). In line with this observation, we found a relative underperformance of juvenile mice in the EBC, linked to anterior cerebellar regions (*Figure 6B₂*).

## Distinct developmental trajectories of specific cerebellar circuitries

PCs are generated in the ventricular zone at the base of the fourth ventricle (*Morales and Hatten, 2006*; *Hoshino et al., 2005*; *Yamada et al., 2014*). Cerebellar compartmentalization has been shown to correlate with PCs birth dates: early-born (E10-E11.5) will become adult Z+ PCs and late-born (E11.5-E13) will become adult Z– PCs (*Hashimoto and Mikoshiba, 2003*; *Namba et al., 2011*; *Larouche and Hawkes, 2006*). Thus, there is a strong relationship between embryonic compartments and adult zonal patterning associated with the modular organization of the olivo-cortico-nuclear circuit (*Brochu et al., 1990*; *Sugihara and Shinoda, 2004*; *Sugihara and Shinoda, 2007a*; *Voogd and Ruigrok, 2004*; *Voogd et al., 2003*; *Pijpers and Ruigrok, 2006*). The maturation of the

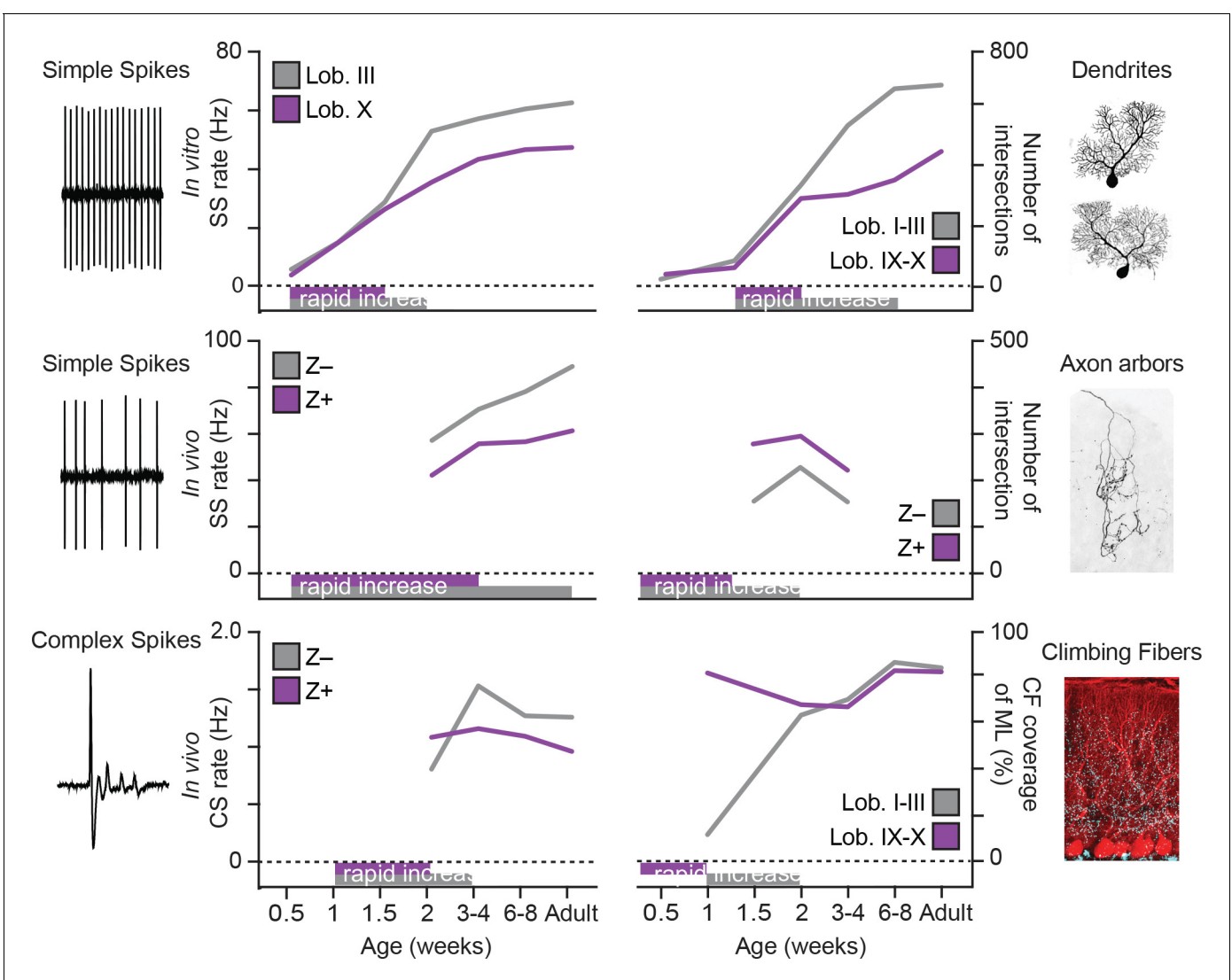

**Figure 7.** Summary of developmental timelines of Purkinje cell activity and morphology. Left, selected features for physiological activity, represented by simple spike firing rate *in vitro*, simple spike rate *in vivo* and complex spike rate *in vivo*. Below the graphs periods of rapid development are indicated, either based on our data or by extrapolation when first recorded levels are already high, see e.g. simple spike rate *in vivo*. Right, selected morphological features, represented by dendritic shape, axonal shape (both as total number of intersections) and climbing fibers (as % of climbing occupation of molecular layer). Note that, although not all data points are available and extrapolation is required, the general trend emerging is that of rapid development in the second postnatal week (P7 to P14) and that Lobule III/ZebrinII– PCs typical complete rapid development later. Z– = ZebrinII negative, Z+ = ZebrinII positive, SS = simple spike, CS = complex spike.

zonal phenotype is only complete around P12-P15, based on the molecular markers ZebrinII (*Brochu et al., 1990*; *Lannoo et al., 1991*) or PLCβ4 (76). Although this patterned organization of heterogeneity is a unique feature of the cerebellum, its impact on early cerebellar circuitry assembly was unknown. In our studies, developmental electrophysiological and morphological analyses revealed that lobule X/Z+ PCs reached adult stable properties earlier than anterior lobule III/Z– PCs (*Figure 7*), particularly when it comes to simple and complex spike activity, as well as dendritic morphology. This suggests that the configuration of mature PCs follows directly from their development, which is a consequence of their birthdate. *In vivo* recordings followed by post-mortem analysis allowed us to group PC acquired throughout the cerebellum into Z+ and Z– subpopulations. Signs of ZebrinII-related differentiation in the electrophysiological properties, similar to those observed in firing rate of SS in the adult mouse *in vivo* (*Zhou et al., 2014*), could already be detected in the end at the second postnatal week from P12 (*Figure 1C$_1$*). When examined across lobules, rather than in direct relation to ZebrinII, the slower development appears most obvious in lobules I-II, III, VI-VII and the Z– PCs in the hemispheres, while particularly lobule X and the flocculus appear to reach adult levels at P18-P29 (*Figure 1—figure supplement 3A–B*), but prohibit us from making conclusive statements. As also found in adult mice based in *in vitro* recording from lobules III and X (*Zhou et al., 2014*; *Wu et al., 2019*), the difference at P12-P19 is, at least in part, caused by differences in intrinsic activity (*Figure 2B$_1$*). We focus our *in vitro* studies of intrinsic activity on lobule III in the anterior vermis and lobule X in the nodular vermis (*Brochu et al., 1990*; *Zhou et al., 2014*; *Sugihara and Shinoda, 2007a*; *Figure 2A$_1$*), as these regions can be readily identified in the cerebellum at all ages, are oriented in such a way that it is possible to record from them in a single sagittal slice and have previously been demonstrated to be representative of the Z– and Z+ PC populations (*Zhou et al., 2014*; *Wu et al., 2019*). Z+ PCs demonstrate early maturation of their physiological properties by reaching a SS activity level *in vivo* at P18 that is not significantly different from that in adulthood. Contrarily, in PCs from anterior regions of the cerebellum the activity continues to gradually increase until adulthood (*Figure 1—figure supplement 3*). Similarly, CS activity, which results exclusively from the activity of neurons in the inferior olive, also showed reduced activity of Z+ PCs compared with Z– PC CSs from P18-P27 (*Figure 1D$_1$*). Overall, physiological differences in two populations of PCs are first detected at the end of the second postnatal week and result from intrinsic differences combined with an input component (*Figure 2—figure supplement 2*).

Previous reports have shown that, in rats, changes in physiological properties during development match the time course of dendritic growth, without differentiating subtypes (*McKay and Turner, 2005*). Indeed, we observed a marked dendritic growth around the second and third week in the mouse. Starting from P18, PCs of the anterior lobule III continue to grow and develop a larger dendritic area and more elaborated dendritic tree when compared with the nodular lobule X PCs. Similar to our physiology data, PCs in the nodular lobule X region appear to reach their adult stage at P18, although a smaller growth is still present from the adolescent to adult stage. This suggests that in some populations of PCs their maturation is not only linked with their location in the cerebellum (*Altman, 1972*), but also coupled to their physiological development. The development of cerebellar circuits is not solely dependent on an intrinsic genetic code that regulates cell autonomous early activity and early morphologic features. During development, different types of afferents target specific regions of the cerebellum rearranging its circuitry. In particular, mossy fiber (*Sotelo and Wassef, 1991*; *Voogd and Ruigrok, 1997*; *Ji and Hawkes, 1995*; *Armstrong et al., 2009*) and CF (*Voogd and Ruigrok, 1997*; *Chédotal et al., 1997*; *Sotelo and Chedotal, 2005*) afferents terminate into parasagittal domains that align with the ZebrinII domains. CF development has been carefully described (*Watanabe and Kano, 2011*; *Hashimoto et al., 2009*; *Kano et al., 2018*). Our results revealed a unique temporal development of the functional differentiation of the CF synapses. Concomitantly with early maturation of the nodular lobule X described previously also CF translocation occurs earlier in this cerebellar region (*Figure 4*). Additionally, CS activity *in vivo* shows that differences between Z– and Z+ PCs are evident by P12 but in the juvenile age group (P18-P29) they resemble the direction observed at adult mature stages. This period, P12-P18, is known to be the late phase of CF elimination (*Kano et al., 2018*). Although PCs during development are innervated by multiple CFs (*Hashimoto et al., 2009*; *Hashimoto and Kano, 2005*), the CS rate has never been found to be higher during development than in adulthood (*Arancillo et al., 2015*; *Sokoloff et al., 2015*; *Kawamura et al., 2013*), and we verify that here for both Z+ and Z– populations. Our data also reveal that CF pause duration is normalized after the period of synapse elimination at P18

(*Figure 1D₂*). Initial dendritic outgrowth is known to be independent of input signals (*Sotelo and Arsenio-Nunes, 1976*; *Alvarado-Mallart and Sotelo, 1982*; *Dusart et al., 1997*). However, CFs are necessary to complete PC dendritic growth and synapse maturation (*Sotelo and Arsenio-Nunes, 1976*; *Sotelo, 2004*). Moreover, positional cues from PCs are necessary for the correct CF-PC connection (*Chédotal et al., 1997*). When during development these molecules are expressed in diverse PCs subtypes could partially explain specific levels of CF maturation. During the second postnatal week features substantial changes in PC axon arbors as well. Complexity and length both significantly increase between P7 and P10 and P14. However, between P14 and P21, morphological pruning appears to take place (*Figure 5*). PC axons in culture also feature a multi-step development, first extending neurites and then switching to an arborizing phase (*de Luca et al., 2009*). However, this development had not been examined *in vivo*. PC axons in the adult feature differences between subnuclei of the cerebellar nuclei (*Sugihara et al., 2009*). While axons within the dentate nucleus exhibit very compact arbors, axons within the medial nucleus are not dense. We were able to detect signatures of this inter-nuclear difference in the developing mouse cerebellum. However, the intra-nuclear differences are subtle in the development of Z– and Z+ axon arbors. The differences between the subpopulations appear to be set early on and growth takes place within that difference already existing.

Developmental patterning of SS and CS rates *in vivo*, as well as that of the axonal projection features, were analyzed across the populations of Z– and Z+ PCs. SS activity *in vitro*, dendritic development and CF innervation were taken from virtually exclusively Z– anterior and Z+ nodular lobules, because ZebrinII is not expressed at the earlier time points tested for these parameters and the need for coronal slicing hinders analyzing multiple time points. The possibility that some of the observed differences are related to lobules or transverse zones cannot be excluded (*Kim et al., 2009*; *Hawkes and Eisenman, 1997*; *Armstrong et al., 2001*), particularly for the morphological changes. However, this selection of lobules is commonly used for comparisons (*Kim et al., 2012*) and have consistently been confirmed with targeted recordings (*Zhou et al., 2014*; *Wu et al., 2019*). Differentiation in the translocation of input(s) further supports the concept that cerebellar circuitries develop at specific and precise timelines that correlate with the subtype of PC and their physical location.

## Spatiotemporal aspects of gene expression patterns and developmental trajectories

There are several lines of evidence that support the concept of spatial regulation of the cerebellum. In mammals, the cerebellar cortex is divided rostrocaudally into four transverse zones (anterior, central, posterior, and nodular) and mediolaterally into parasagittal stripes (*Hawkes and Eisenman, 1997*; *Ozol et al., 1999*; *Armstrong and Hawkes, 2000*; *Armstrong et al., 2000*). Both transverse and parasagittal compartments can be distinguished by gene expression. Analyses of the *meander tail* (*Ross et al., 1990*) and *lurcher* (*Tano et al., 1992*) mutant mice have revealed a genetic compartmental boundary between nodular and anterior lobules. Consistent with this notion, a recent single-cell RNA-sequencing study has shown gene expression correlated or anti-correlated with the ZebrinII pattern (*Rodriques et al., 2019*). A second spatial gene expression pattern was identified related to the vestibulocerebellar region (lobules IX and X), in which genes are exclusively expressed within or outside this region (*Rodriques et al., 2019*). But what is the developmental profile of these spatial expression differences, and what drives the differentiation? Studies suggest that specification of PC subpopulations (both transversal and parasagittal subtypes of PCs) is regulated by cell autonomous mechanisms, more so than activity or afferent dependent, starting from the birth of the PCs (*Hashimoto and Mikoshiba, 2003*; *Namba et al., 2011*; *Larouche and Hawkes, 2006*; *Chung et al., 2008*). *Hashimoto and Mikoshiba, 2003* PC birthdating experiments revealed that, in contrast with the expression of the markers available at the time, the eight clusters of PCs the authors identified embryonically were unchanged until adulthood. Although ZebrinII is a 'late-onset' parasagittal marker, others, such as neurogranin (*Larouche et al., 2006*), were shown to be expressed from E14.5 in a pattern that was maintained until P20. While *Slc1a6* expression is similar to that of ZebrinII in adult mice, its mRNA can already by detected from embryonic ages and immunohistochemical analysis supports a patterned, selective expression in the caudal cerebellum from embryonic day 18 (*Yamada et al., 1997*). Additionally, PLCβ4 expression is restricted to the Z– cells and its expression also begins just before birth. The PLCβ4-positive clusters in the neonate are

complementary with the Z− neurons (*Marzban et al., 2007*). With the use of mouse genetics (*Sillitoe et al., 2009*) or adenovirus tracing (*Namba et al., 2011*) it was possible to verify that the birth date-related PCs correlated with the zonal pattern of ZebrinII. Finally, it is known that the nodulus (vermal lobule X) and the flocculus are distinct divisions of the cerebellum at an early age (*Fujita et al., 2012*). Our electrophysiological data add to this notion that differences in physiological properties result from intrinsic properties of PCs (*Figure 2*) early in development. Intrinsic activity could be observed as early as P3 and differences between PC subpopulations become apparent starting from P12. Moreover, we observed that the influence of extrinsic inputs, the net effect of excitatory and inhibitory inputs, is relatively small in the second week, but starts to drive activity at P18. While this influence remains relatively stable in nodular lobule X/Z+ PCs, the net driving effect of extrinsic inputs increases over time for anterior lobule III/Z− PCs until the cells reach their full maturation in the adult (*Figure 2—figure supplement 2*). It should be noted, however, that the methods we employed do not allow us to determine the individual contributions of each input, except that of the CFs, and that this maturation could also be the result of a shift in the balance between excitation and inhibition (*Jelitai et al., 2016*).

While the precise molecular mechanism underlying these differences is not well-defined, several lines of evidence suggest gene expression differences in subtypes of PCs may explain this phenomenon. For instance, Z− PCs have a higher expression of TRPC3 (44) and its ablation decreases their firing rate to make them more similar to the firing rate of Z+ PCs with behavioral consequences. Additionally, the TRPC3 channel molecular cascade includes proteins expressed in parasagittal bands such as mGluR1b (*Mateos et al., 2000*), IP3R1 (*Furutama et al., 2010*), PLCβ3/4 (*Sarna et al., 2006*), and PKCδ (*Barmack et al., 2000*). Recently, specific ATPases and potassium channels were identified in Z− PCs that can also contribute to explain intrinsic differences (*Rodriques et al., 2019*). The physiology and morphology of PCs have striking interaction effects. As described previously (*McKay and Turner, 2005*; *Bradley and Berry, 1979*), from the second postnatal week PCs develop their characteristic dendritic tree coupled with functional transitions (*Dusart and Flamant, 2012*). Our data show for the first time that coupled with their electrophysiological properties, nodular lobule X PCs reach maturity of their dendritic tree faster than anterior lobule III PCs (*Figure 3*).

During the second postnatal week of development also the complexity of the PC axonal arbor increases significantly (*Figure 5*). Corticonuclear topography is related with the topography of the olivocerebellar pathway (*Sugihara et al., 2009*; *Sugihara and Shinoda, 2007a*), in that Z+ PCs typically project to the lateral/caudoventral cerebellar nuclei while Z− neurons typically project to the medial/rostrodorsal parts of the cerebellar nuclei. In adult rats, PC axon arbors have been shown to vary morphologically based on their location within the cerebellar nuclei with more Z+ subnuclei containing denser, more complex PC axonal arbors (*Sugihara et al., 2009*). Our data shows for the first time the existence of a similar pattern of distinct morphology of PC axons in the different subnuclei during development (*Figure 5*). Although neuronal subtypes of the related vestibular nuclei have been identified (*Shin et al., 2011*), neither the subtypes in the cerebellar nuclei nor the pattern of PC to cerebellar nuclei neuron projections have been comprehensively studied. Linking the projection pattern of Z+ and Z− PCs to genetically identified neuronal subpopulations of each cerebellar nuclei is a crucial future step in understanding the development and functioning of the olivocerebellar circuit.

## Differential ontogeny of cerebellar-sensorimotor functions

Unique timelines in the maturation of cerebellar microcircuitries sparked the hypothesis that correlated behaviors are impacted and have distinct developmental profiles as well. Although the list of cerebellum-related behaviors is long, only few tasks have been directly linked to specific, restricted cerebellar regions. The difference in developmental timeline of PC SS activity, which directly influences downstream targets, is comparable between flocculus and nodulus, as well as between Z− PCs in the hemispheres and anterior cerebellum (*Figure 1—figure supplement 3B*). Therefore, we examined VOR adaptation and EBC as proxies for Z+ and Z− cerebellar module-related behaviors, respectively. VOR adaptation and EBC are regulated by different modules in the cerebellum: VOR adaptation is controlled by the flocculus of the vestibulocerebellum, which is a Z+ region (*Zhou et al., 2014*; *Sugihara and Quy, 2007b*; *Lisberger, 1988*; *Ito, 2002*; *Fujita et al., 2014*), and EBC is controlled by the hemispheric lobule VI, which is predominantly a Z− region (*Mostofi et al.,*

*2010*; *Boele et al., 2010*; *Thompson and Steinmetz, 2009*; *Heiney et al., 2014*; *Hesslow, 1994a*; *Hesslow, 1994b*). Our data show that different elements of the nodular Z+ circuitry mature, that is reach their adult levels at the juvenile (P21) stage, earlier than anterior Z– circuitries and hence we hypothesize that this differentiation would manifest itself in the ability to perform related learning tasks. We found that young animals, compared to adult mice, start with a similar OKR baseline similar, but a lower VOR baseline (*Figure 6—figure supplement 1*). A similar attenuation of VOR gain, but more specifically for lower to midrange frequencies, was observed previously, while we were not able to replicate the higher OKR gain at 1.0 Hz visual stimulation observed in that work (*Faulstich et al., 2004*). Taken together, both studies suggest a delayed maturation of the VOR with a potential compensatory role for an 'overactive' OKR system. As OKR gain is attenuated by loss or dysfunctional cerebellar input, the VOR gain typically increases in those conditions (*Schonewille et al., 2010*; *van Alphen and De Zeeuw, 2002*) and hence these results argue against any impairments due to incomplete development of the floccular region. When challenged with a learning paradigm, the VOR phase reversal, the younger animals adapted faster than adult animals. These results support our hypothesis that the flocculus circuitry at P21 is functional and suggests that it even allows for faster learning, a form of immature hyperplasticity, comparable to known forms of enhanced neural plasticity in development (*Wiesel and Hubel, 1965*; *Hensch and Bilimoria, 2012*; *Cai et al., 2014*). Additionally, when comparing the simple spike rate development in the flocculus with the nodular regions of the cerebellum (*Figure 6—figure supplement 3B₂*), even with limited data points, data suggests that by P18-29 these PCs have reached their adult firing rate stage. In contrast, younger animals have a poorer performance in the CR of the EBC paradigm compared to adult animals. Functional immaturity of the eyeblink response has been suggested to be due to immaturity of the afferent pathway when using an auditory cue (*Nicholson and Freeman, 2000*). However, the fact that we use a visual cue and compare the results with visual stimulus-driven OKR and VOR adaptation, argue against the role of an immature afferent pathway, while the presence of normal URs rejects the premise that an inability to blink is the cause. At P21 the firing rate of anterior Z– PCs in the eyeblink region (*Figure 1—figure supplement 3A₃,B₁*), is reduced when compared to adult counterparts (*Figure 1—figure supplement 3A₄,B₁*), which can likely contribute to the impaired CR in young animals (*Wu et al., 2019*; *ten Brinke et al., 2015*). Evidence suggests that Z– and Z+ PCs utilize different forms of synaptic plasticity, but there is still no consensus on which plasticity mechanisms underlie VOR adaptation and EBC. Long-term depression (LTD) was the first type of synaptic plasticity implicated in cerebellar motor learning (*Ito, 2000*). LTD has been suggested to participate in the EBC response (*Grasselli and Hansel, 2014*; *Freeman, 2015*) and this form of plasticity is known to occur on Z– lobule III PCs while in Z+ lobule X PCs is not detected (*Paukert et al., 2010*; *Wadiche and Jahr, 2005*). Interestingly, blocking TRPC3 function eliminates LTD plasticity (*Kim, 2013*) and, we recently showed that, TRPC3 loss-of-function in mice showed an impaired EBC response (*Wu et al., 2019*), but normal VOR adaptation. LTD can be readily induced in the anterior regions of juvenile PCs (*Kim, 2013*) and LTD-deficient mice do not have impaired EBC, arguing against a central role for LTD in the EBC impairment in juvenile mice. Intrinsic excitability is increased after EBC (*Schreurs et al., 1997*; *Titley et al., 2020*) and deletion of calcium-activated potassium channel SK2 ablated this plasticity of intrinsic excitability (intrinsic plasticity) and resulted in impaired EBC, but enhanced VOR adaptation (*Grasselli et al., 2020*), a phenotype that is in line with the juvenile phenotype (*Hesslow, 1994b*). Thus, a lack of ability to reach higher levels of excitability by young anterior Z– PCs could explain the lower performance in the P21 animals. The latter example suggests that the temporal difference in the emergence of cerebellum-sensorimotor behaviors (*Figure 6*), could result from a distinct cell-autonomous excitability regulation in distinct PCs populations. Future studies will have to determine the precise parameters for induction of the difference forms of plasticity in each region, and their respective timelines.

Overall, this study highlights the heterogeneity within the cerebellum during development. Key parameters such as CF input, dendritic and axonal shape and intrinsic firing rate reach mature levels at different moments in postnatal development, depending on the subtype of PC and the regional location. The evolutionary advantage to have distinct developmental timelines in different cerebellar regions is likely to be related with their function. Increasing evidence has shown that in addition to sensorimotor processing the cerebellum has a role in cognitive functions (*White and Sillitoe, 2013*; *Ito, 2008*) and early cerebellar dysfunction has been implicated in neurodevelopmental disorders (*Wang et al., 2014*; *Martin and Albers, 1995*; *Kern, 2002*). Recently, a comprehensive study has

shown that the diverse motor and non-motor functions of the cerebellar vermis are mediated by different groups of fastigial output neurons with specific connections. The authors identified two major classes of fastigial glutamatergic projection neurons: small neurons innervated by Z+ PCs, projecting to circuits associated with sensory processing, motor preparation and behavioral, cognitive, affective and arousal responses to novel or unexpected events, while large neurons innervated by Z– PCs connect with circuits associated with control of motor and autonomic functions (*Fujita et al., 2020*). Thus, uncovering the mechanisms underlying early circuitry formation in the developing cerebellum is imperative to understand the basis of cerebellum circuitry and associated disorders. Additional experiments are necessary to clarify developmental stages of other elements of cerebellar circuits such as mossy fiber or interneurons. Nevertheless, our results demonstrate that the emergence of cerebellar sensorimotor functions are tightly coupled with distinctive PC properties.

# Materials and methods

## Key resources table

| Reagent type (species) or resource | Designation | Source or reference | Identifiers | Additional information |
|---|---|---|---|---|
| Strain, strain background (*Mus musculus*) | *Slc1a6-EGFP* | **Gong et al., 2003** | MMRRC: 012845-UCD | (*Tg(Slc1a6-EGFP) HD185Gsat/Mmucd*) |
| Strain, strain background (*Mus musculus*) | *Pcp2-cre*$^{ERT2}$ | Institut Clinique de la Souris, France Jackson Laboratory | MGI:97508; ICS: 0273 | (*Tg(Pcp2-creER$^{T2}$)17.8.ICS*) |
| Strain, strain background (*Mus musculus*) | *Ai14* | The Jackson Laboratory | JAX: 007908 | (*Gt(ROSA)26Sor$^{tm14(CAG-tdTomato)Hze}$/J*) |
| Strain, strain background (*Mus musculus*) | C57BL/6 | The Jackson Laboratory Janvier Labs Charles River | JAX: 00055 Janvier: C57BL/6JRj CR: C57BL/6NCrl | |
| Antibody | Goat anti-ZebrinII/ Aldolase C (Goat polyclonal) | Santa Cruz Biotechnology | Cat# sc-12065 RRID:AB_2242641 RRID:AB_2315622 | 1:1000 |
| Antibody | Mouse anti- Calbindin D-28K (Mouse monoclonal) | Swant | Cat# 300 RRID:AB_10000347 | 1:10,000 |
| Antibody | Guinea pig anti-VGluT2 (Guinea pig polyclonal) | Millipore | Cat# AB2251-I RRID:AB_2665454 | 1:2000 |
| Antibody | Rabbit anti-RFP (Rabbit polyclonal) | Rockland | Cat# 600-401-379 RRID:AB_2209751 | 1:1000 |
| Antibody | Cy3 Streptavidin | Jackson ImmunoResearch | Cat# 016-160-084 RRID:AB_233724 | 1:1000 |
| Antibody | Cy3-AffiniPure Donkey anti-Mouse (Mouse polyclonal) | Jackson ImmunoResearch | Cat# 715-165-150 RRID:AB_2340813 | 1:1000 |
| Antibody | Alexa Fluor 488-AffiniPure Donkey anti-Guinea Pig (Guinea pig polyclonal) | Jackson ImmunoResearch | Cat# 706-545-148 RRID:AB_2340472 | 1:1000 |
| Antibody | Donkey anti-goat Daylight 488 (Goat polyclonal) | Jackson ImmunoResearch | Cat# 705-486-147 RRID:AB_2616594 | 1:500 |
| Antibody | Cy3-AffiniPure Donkey Anti-Rabbit (Rabbit polyclonal) | Jackson ImmunoResearch | Cat# 711-165-152 RRID:AB_2307443 | 1:500 |
| Chemical compound, drug | 4′,6-Diamidine-2′-phenylindole dihydrochloride (DAPI) | Thermo Fisher Scientific | Cat# D3571 RRID:AB_2307455 | |
| Chemical compound, drug | Paraformaldehyde (PFA) | Millipore | Cat# 104005 | 4% |
| Chemical compound, drug | Biocytin | Sigma-Aldrich | Cat# B4261 | 1% |
| Chemical compound, drug | Evans Blue | Sigma-Aldrich | Cat# E2129 | 0.5% |
| Chemical compound, drug | Picrotoxin | Hello Bio Ltd | Cat# HB0506 | 100 µM |

*Continued on next page*

*Continued*

| Reagent type (species) or resource | Designation | Source or reference | Identifiers | Additional information |
|---|---|---|---|---|
| Chemical compound, drug | NBQX | Hello Bio Ltd | Cat# HB0442 | 10 µM |
| Chemical compound, drug | D-AP5 | Hello Bio Ltd | Cat# HB0225 | 50 µM |
| Chemical compound, drug | Mineral oil | Sigma-Aldrich | Cat# M3516 | |
| Chemical compound, drug | Tamoxifen | Sigma-Aldrich | Cat# T5648 | |
| Software, algorithm | FIJI (ImageJ) | National Institute of Health | RRID:SCR_002285 | |
| Software, algorithm | MATLAB 2008, 2016 | MathWorks | RRID:SCR_001622 | |
| Software, algorithm | ZEN digital Imaging for Light Microscopy | ZEISS | RRID:SCR_013672 | |
| Software, algorithm | Leica Application Suite X (LAS X) | Leica Microsystems | RRID:SCR_013673 | |
| Software, algorithm | Clampfit 10 | Molecular Devices | RRID:SCR_011323 | |
| Software, algorithm | Patchmaster | HEKA Electronics | RRID:SCR_000034 | |
| Software, algorithm | GraphPad Prism | GraphPad Software | RRID:SCR_002798 | |
| Software, algorithm | Compensatory eye movements analysis | Schonewille group | | https://github.com/MSchonewille/iMove |
| Software, algorithm | Eyeblink conditioning analysis | Neurasmus B.V. Rotterdam | RRID:SCR_021043 | |
| Other | Leica SM2000 R sliding microtome | Leica Biosystems | RRID:SCR_018456 | |
| Other | P-1000 Puller | Sutter Instrument | RRID:SCR_021042 | |
| Other | LSM 700 laser scanning confocal | ZEISS | RRID:SCR_017377 | |
| Other | SP5 confocal | Leica Microsystems | RRID:SCR_020233 | |
| Other | SP8 confocal | Leica Microsystems | RRID:SCR_018169 | |
| Other | Axio Imager.M2 | ZEISS | RRID:SCR_018876 | |

## Mice

All animals in this study were handled and kept under conditions that respected the guidelines of the Dutch Ethical Committee for animal experiments and were in accordance with the Institutional Animal Care and Use Committee of Erasmus MC (IACUC Erasmus MC), the European and the Dutch National Legislation. All animals were maintained under standard, temperature controlled, laboratory conditions. Mice were kept on a 12:12 light/dark cycle and received water and food *ad libitum*. The following transgenic mouse lines were used in this study: *Slc1a64-EGFP* (*Tg(Slc1a6-EGFP) HD185Gsat/Mmucd*) (**Gong et al., 2003**), *Pcp2-cre$^{ERT2}$* (*Tg(Pcp2-creER$^{T2}$)17.8.ICS*) (**Wu et al., 2019**), and *Ai14* (*B6;129S6-Gt(ROSA)26Sor$^{tm14(CAG-tdTomato)Hze}$/J*) (**Madisen et al., 2010**). The following primer sequences were used for routine genotyping: *Slc1a6-EGFP* (5′-TTCCTGATTGCTGGAAAGATTCTGG −3′; 5′-AGTTCAGGGAAAGGCCA TACCTTGG-3′; 5′-GGATCGGCCATTGAACAAGATGG-3′; 5′-AAGTTCATCTGCACCACCG-3′; 5′-TCCTTGAAGAAGATG GTGCG-3′), *Pcp2-cre$^{ERT2}$* (5′-CCA TGGTGATACAAGGGACATCTTCC-3′; 5′-CATGTGAAATTGTGCTG CAGGCAGG-3′; 5′-GCTATGAC TGGGCACAACAGACAATC-3′; 5′-CAAGGTGAGATGACAGGAGATC CTG-3′), and *Ai14* (5′-CTG TTCCTGTACGGCATGG-3′; 5′-CCGAAAATCTGTGGGAAGTC-3′; 5′-GGCATTAAAGCAGCGTATCC-3′; 5′-AAGGGAGCTGCAGTGGAGTA-3′). Both male and female mice were used in all experiments.

## *In vivo* extracellular recordings and analysis

The *in vivo* extracellular recordings were performed in a total of 161 mice with an age range from P12 to P269. We used either *Slc1a6-EGFP* (**Dehnes et al., 1998**) or C57BL/6J mice to record PCs. Briefly, mice were maintained under general anesthesia with isoflurane/O$_2$ (4% induction and 1–5–2% maintenance) while five holes were drilled using a high-speed diamond-tipped drill (Foredome, Bethel, CT, USA, RRID:SCR_021046). To obtain electrocorticogram (ECoG) signals, five pure silver

ball-tipped electrodes (custom-made from 0.125 mm diameter silver wire; Advent research materials LTD, Eynsham, Oxford, United Kingdom, RRID:SCR_021045) were placed on the meningeal layer of the dura mater. Two silver electrodes were positioned bilateral above the primary cortex (M1, 1 mm rostral; 1 mm lateral; relative to Bregma), two were placed above the primary sensory cortex (S1, 1 mm caudal; 3.5 mm lateral; relative to Bregma), and one in the interparietal bone (1 mm caudal; 1 mm lateral; relative to Lambda). UV-sensitive composites, a layer of Optibond (Kerr, Bioggio, Switzerland) and Charisma Flow (Heraeus Kulzer, Hesse, Germany), were used to fix the silver electrodes and a pedestal in the mouse head. To obtain extracellular recordings a craniotomy was made in the occipital bone and temporarily closed with Kwik-Cast sealant (World Precision Instruments Inc, Sarasota, FL, USA, RRID:SCR_008593) to prevent cooling of the brain. In the end of the surgery, mice received 0.1–0.2 ml saline for hydration and 0.2 l O2/min. ECoG and extracellular recordings were sampled at 20 kHz (setup 1: Digidata 1322A, Molecular Devices LLC., Axon instruments, Sunnyvale, CA, USA, RRID:SCR_021041), amplified, and stored for offline analysis (CyberAmp and Multiclamp 700A, Molecular Devices, RRID:SCR_021040) or at 50 kHz (setup 2: ECoG: adapted MEA60, Multichannel system, Reutlingen, Germany, RRID:SCR_021039; extracellular: Multiclamp 700B amplifier, RRID:SCR_018455, with a Digidata 1440; Molecular Devices, RRID:SCR_021038). Single-unit recordings started two hours after the termination of isoflurane application, only when the ECoG looked normal for an active mouse in an alert status. We recorded using borosilicate glass pipettes (Harvard apparatus, Holliston, MA, USA, RRIDSCR_021037) with 0.5–1.0 µm tips and a resistance of 6–12 MΩ. Glass pipettes were filled with internal solution containing (in mM): 9 KCl, 3.48 MgCl2, 4 NaCl, 120 K+-Gluconate, 10 HEPES, 28.5 Sucrose, 4 Na2ATP, 0.4 Na3GTP in total pH 7.25–7.35, osmolarity 290–300 mOsmol/Kg (Sigma-Aldrich, Merck KGaA, Darmstadt, Germany, RRID:SCR_008988); and 1% biocytin or 0.5% Evans Blue. At the recording location biocytin was released with iontophoresis with 1 s pulses of 4 µA for 3 min (custom-built device, Erasmus MC, Rotterdam, The Netherlands, RRID:SCR_002737) or Evans blue was injected with pressure. This procedure was done to identify the location of the recordings. In our analysis, we included only the cells that we could identify the recording location by the use of the injection spot. For spike analysis of the PCs, only cells with a recording length of at least 90 s were included in the study (duration: 214 ± 160 s). All *in vivo* recordings were analyzed using a MATLAB (MathWorks, Natick, MA, USA, RRID:SCR_001622) code to detect spikes using threshold and principal component analysis (*Aminov et al., 2012*) and a custom build MATLAB code to analyze inter spike variables. The CV is the variation in inter-spike-intervals (ISI) during firing and was calculated by dividing the standard deviation by the mean of ISIs. The CV2 represents the variance on a spike-to-spike base, it is less sensitive for a single outlier and was calculated as 2*|ISIn + 1-ISIn| / (ISIn + 1 + ISIn). The regularity index was calculated by extracting regular spike patterns, using a CV2 threshold of <0.2 for at least three consecutive spikes (*Shin et al., 2007*). Adult >P60 PCs data set localized in the flocculus used in this study has also been used in a previous study (*Zhou et al., 2014*).

### *In vitro* extracellular recordings and analysis

The *in vitro* extracellular recordings were performed in a total of 49 mice with an age range from P3 to P378. We used either *Slc1a6-EGFP* (*Gong et al., 2003*; *Dehnes et al., 1998*) or C57BL/6J mice to record PCs. As previously described (*Wu et al., 2019*), the brain was quickly removed and placed in ice-cold slice solution (continuously carbogenated with 95% O2 and 5% CO2) containing the following (in mM): 240 Sucrose, 2.5 KCl, 1.25 NaH2PO4, 2 MgSO4, 1 CaCl2, 26 NaHCO3, 10 D-glucose. Acute sagittal slices 250 µm thick of vermal cerebellar tissue were cut in ice-cold slicing solution using a vibratome (VT1000S, Leica Biosystems, Wetzlar, Germany, RRID:SCR_016495) with a ceramic blade (Campden Instruments Ltd, Manchester, United Kingdom, RRID:SCR_021036). Directly after slicing, the slices were transferred to a recovery bath and were incubated in oxygenated artificial cerebrospinal fluid (ACSF) and maintained at 34°C for one hour. The ACSF was continuously carbogenated with 95% O2 and 5% CO2 and consisted of (in mM): 124 NaCl, 5 KCl, 1.25 Na2HPO4, 2 MgSO4, 2 CaCl2, 26 NaHCO3, 20 D-glucose. After incubation period, slices were transferred to room temperature. To record the individual slices, these were transferred to a recording chamber and maintained at 34 ± 1°C with a feedback temperature controller with heater (Scientifica, Uckfield, United Kingdom, RRID:SCR_021035) under continuous superfusion with the oxygenated ACSF.

For all the recordings, slices were bathed with ACSF supplemented with synaptic receptor blockers, NMDA receptor antagonist D-AP5 (50 µM, Hello Bio Ltd, Bristol, United Kingdom, RRID:SCR_

021047), selective and competitive AMPA receptor antagonist NBQX (10 µM, Hello Bio Ltd, Bristol, United Kingdom, RRID:SCR_021047), non-competitive GABA$_A$ receptor antagonist and glycine receptor inhibitor Picrotoxin (100 µM, Hello Bio Ltd, Bristol, United Kingdom, RRID:SCR_021047). PCs were visualized with SliceScope Pro 3000, a CCD camera, a trinocular eyepiece (Scientifica, Uckfield, United Kingdom, RRID:SCR_021035) and ocular (Teledyne Qimaging, Surrey, Canada). Whole-cell and cell attached recordings were obtained using borosilicate pipettes (Harvard apparatus, Holliston, MA, USA RRIDSCR_021037) with a resistance of 4–6 MΩ, filled with internal solution containing (in mM): 9 KCl, 3.48 MgCl$_2$, 4 NaCl, 120 K$^+$-Gluconate, 10 HEPES, 28.5 Sucrose, 4 Na$_2$ATP, 0.4 Na$_3$GTP in total pH 7.25–7.35, osmolarity 290–300 mOsmol/Kg (Sigma-Aldrich, Merck KGaA, Darmstadt, Germany, RRID:SCR_008988). Cell-attached recordings were made with a seal of 30 MΩ to 2 GΩ and lasted for a minimum of 90 s up to 150 s. Recording pipettes were supplemented with 1 mg/ml biocytin to allow histological staining to identify PCs location. Cell-attached recordings were performed using an ECP-10 amplifier (HEKA Electronics, Lambrecht, Germany, RRID:SCR_018399) and digitized at 20 kHz. Acquisition was done in Patchmaster (HEKA Electronics, Lambrecht, Germany, RRID:SCR000034) and ABF Utility (Synaptosoft, Fort Lee, NJ, USA, RRID:SCR_019222) was used to convert the Patchmaster files for analysis. Clampfit 10 (Molecular Devices, LLC, San Jose, USA, RRID:SCR_011323) was used to analyze spikes (*Aminov et al., 2012*) and a custom-build MATLAB code (MathWorks, Natick, MA, USA, RRID:SCR_001622) using inter spike properties was used to analyze spike variables (*Shin et al., 2007*).

## Immunohistochemistry

For immunohistochemistry, mice were deeply anesthetized with sodium pentobarbital, perfused transcardially with sodium chloride solution (Baxter International Inc, Deerfield, IL, USA, #TKF7124, RRID:SCR_003974) followed by 4% paraformaldehyde (PFA) in 0.1M phosphate buffer (PB). The dissected brains were then post-fixed in 4% PFA for 2 hr at 4˚C and then cryoprotected in 10% sucrose/0.1M PB (Sigma-Aldrich, #S-0389, RRID:SCR_008988) overnight at 4˚C. Next day, the brains were embedded in 14% gelatin/30% sucrose/0.1M PB solution (gelatin: FujiFilm Wako Pure Chemical Corporation, Osaka, Japan, #077–03155, RRID:SCR_021034), fixed for 2 hr at room temperature and incubated overnight in 30% sucrose/0.1M PB solution.

For the *in vivo* electrophysiology, the injection spot was identified by the presence of biocytin or Evans blue. Briefly, brains were sectioned 40–100 µm thick in a coronal plane with a freezing microtome. Free-floating sections were rinsed with 0.1M PB and blocked for 2 hr in a solution of 0.5% Triton X-100/10% normal horse serum/0.1M PB at room temperature. With the exception of *Slc1a6-EGFP*-positive sections, all sections were incubated 4 days at 4˚C in a solution of 0.5% Triton X-100/2% normal horse serum/0.1M PB with primary antibody against Aldolase C (1:1000, goat polyclonal, Santa Cruz Biotechnology, Dallas, TX, USA RRID:AB_2242641). After rinsing the sections with 0.1M PB, sections were incubated 2 hr at room temperature in a solution of 0.5% Triton X-100/2% normal horse serum/0.1M PB with secondary antibody Alexa Fluor 488-AffiniPure Donkey anti-goat (1:500, Jackson Immuno Research Labs, West Grove, PA, USA, RRID:AB_2340428) and Cy3-streptavidin (1:1000, Jackson Immuno Research Labs, RRID:AB_2337244). Finally, brain slices were incubated for 10 min with DAPI (Thermo Fisher Scientific, Waltham, MA, USA, RRID:AB_2629482) in 0.1M PB, rinsed with PB, mounted in slides in chrome alum (gelatin/chromate) and mounted with Mowiol (Polysciences Inc, Warrington, PA, USA, #17951).

To label recorded PCs, free-floating sagittal brain slices 250 µm thick obtained from *in vitro* electrophysiology recordings were fixed in 4% PFA/0.1M PB overnight at 4˚C. Next, slices were permeabilized with 0.5% Triton X-100 in PB overnight at 4˚C. The following day, brain slices were incubated with Cy3 Streptavidin for 1 hr at room temperature. Finally, brain slices were incubated for 10 min with DAPI in PB, rinsed with PB, mounted in slides in chrome alum (gelatin/chromate) and mounted with Mowiol.

Free-floating sagittal brain slices from P7, P14, P21, P35, and P60 C57BL/6J mice were permeabilized with 0.5% Triton X-100 in 0.1M PB for 1 hr at room temperature and blocked for 2 hr in a solution of 0.5% Triton X-100/10% normal horse serum/0.1M PB at room temperature. After, sections were incubated overnight at 4˚C in a solution of 0.5% Triton X-100/2% normal horse serum/0.1M PB with primary antibody against Calbindin D-28k (1:10000, Swant, Marly, Switzerland, RRID:AB_2314070) and VGluT2 (1:2000, Millipore, RRID:SCR_008983). Next day, sections were rinsed in 0.1M PB and incubated at room temperature for 2 hr in a solution of 0.5% Triton X-100/2% normal horse

serum/0.1M PB with secondary antibodies: Cy3-AffiniPure Donkey anti-Mouse (1:1000, Jackson Immuno Research Labs, RRID:AB_2340813) and Alexa Fluor 488-AffiniPure Donkey anti-Guinea Pig (1:1000, Jackson Immuno Research Labs, RRID:AB_2340472). Finally, brain slices were incubated for 10 min with DAPI in 0.1M PB, rinsed with PB, mounted in slides in chrome alum (gelatin/chromate) and mounted with Mowiol.

To label axonal projections from *Pcp2-cre^{ERT2}; Ai14* mice brains were sectioned 150 µm thick in a coronal plane with a freezing microtome. Free-floating sections were rinsed with 0.1M PB and blocked for 2 hr in a solution of 0.5% Triton X-100/10% normal horse serum/0.1M PB at room temperature. Sections were then incubated overnight at 4°C in a solution of 0.5% Triton X-100/2% normal horse serum/0.1M PB with primary antibodies against Aldolase C (1:1000, goat polyclonal) and RFP (1:1000, Rockland). After rinsing the sections with 0.1M PB, sections were incubated 2 hr at room temperature in a solution of 0.5% Triton X-100/2% normal horse serum/0.1M PB with secondary antibodies Alexa Fluor 488-AffiniPure Donkey anti-goat (1:500, Jackson Immuno Research Labs, RRID:AB_2616594) and Cy3 AffiniPure Donkey anti-rabbit (1:500, Jackson Immuno Research Labs, RRID:AB_2307443). Finally, brain slices were incubated for 10 min with DAPI in 0.1M PB, rinsed with PB, mounted in slides in chrome alum (gelatin/chromate) and mounted with Mowiol.

## Image acquisition and morphological analysis

Images were acquired at 8-bit depth and 1024 × 1024 pixel resolution with Axio Imager.M2 (Carl Zeiss Microscopy, LLC, USA, RRID:SCR_011876), LSM 700 (Carl Zeiss Microscopy, LLC, USA, RRID:SCR_017377) or a SP5/SP8 (Leica Microsystems, Wetzlar, Germany, SP5 RRID:SCR_020233, SP8 RRID:SCR_018169) confocal laser scanning microscope. For each experiment, images were acquired using the same laser power and detection filter settings.

To localize the injection site from *in vivo* recordings, wide-field fluorescent tile scan images were acquired with a 10X objective, 20% overlap and online-stitched. Only recorded neurons with an injection spot small enough to locate the cell to a lobule and its ZebrinII identity were analyzed, other PCs were discarded.

To image individual PCs, images were acquired with 10X/0.3 or 20X/0.8 objectives according with the cell size and with a z-interval of 1 µm. To quantify the dendritic arborization of biocytin-filled PC, the maximum projection of z-stack images of single cells were analyzed with the sholl analysis macro implemented in FIJI (ImageJ, RRID:SCR_002285) software. To quantify the area of a PC, the maximum projection of each image was thresholded in FIJI to fit the area of the cell and measured.

Tile scan images of PCs and CFs from different lobules of the cerebellum were acquired with a 40X/1.3 (Magnification/Numerical Aperture) oil objective with a z-interval of 0.8 µm. For each age group, four sections were imaged (10 images per section) for each brain (three brains per age). For all the following quantifications, the maximum projection of the z-stack was used. Analysis of the VGluT2 synaptic puncta was performed in sagittal sections in regions of lobules I, II, III, IX, and X. Analysis of synapse densities was performed using the 'Analyze Particles' tool in FIJI software to quantify the number of VGluT2 puncta in a region of interest (ROI) and this number was then divided by the area (µm$^2$) of the ROI in the cerebellar cortex. The heights of ML and CFs were measured using three measurements per image. The ML height was measured as the distance from the edge of the PC soma to the apical edge of the ML and the CF height was measured as the distance from the from the edge of the PC soma to the apical edge of the last VGluT2 puncta. The CF extension was quantified as the ratio of the mean of CF height per the mean of ML height per image. Measurements for each mouse were averaged and the numbers computed from each group were pooled and averaged again to obtain the mean of all the measurements made. These analyses were done P7, P14, P21, P35, and P60 C57BL/6J mice and presented as percentages.

Images of PC axon terminal arbors in the cerebellar nuclei were acquired with a 20x objective with a z-interval of 1 µm. For each age group, sections were imaged for each brain (at least three brains per age). Isolated axon arbors were imaged and analyzed with the sholl analysis macro implemented in FIJI software. To quantify the area of a PC, the maximum projection of each image was thresholded in FIJI to fit the area of the cell and measured.

## Compensatory eye movement recordings

Young (P21 on the first day of training) and adult (10–11 weeks old) C57BL/6J mice were used to perform compensatory eye movement recordings, which were described in detail previously (*Schonewille et al., 2010*). Briefly, to head restrain mice during the eye movement recordings a metal construct, *pedestal*, was placed on their skull under general anesthesia with isoflurane/O$_2$. After 3 days of recovery from the surgery, mice were head-fixed and placed in a mouse holder in the center of a turntable (diameter: 60 cm), surrounded by a cylindrical screen (diameter 63 cm) with a random-dotted pattern (*drum*). Compensatory eye movements [optokinetic reflex (OKR), visual vestibular ocular reflex in the light (VVOR) and dark (VOR)] were induced using a sinusoidal rotation of the drum in light (OKR), rotation of the table in the dark (VOR) or the rotation of the table in the light (VVOR) with an amplitude of 5° at 0.1–1 Hz. Motor performance in response to these stimulations was evaluated by calculating the gain (fitted eye velocity/fitted stimulus velocity) and phase (eye to stimulus difference in degrees) of the response.

To study motor learning, mice were subjected to a mismatch between visual and vestibular input to adapt the VOR. The VOR phase-reversal test was done during a period of 5 days, consisting of six 5-min training sessions every day with VOR recordings before, between, and after the training sessions. Between recording sessions, mice were kept in the dark to avoid unlearning of the adapted responses.

The first training day, in-phase stimulation of the visual (the drum) and vestibular stimuli (turntable) rotated in phase at 0.6 Hz and both with an amplitude of 5°, inducing a decrease of gain. In the following days, the drum amplitude was increased to 7.5° (day 2) and 10° (days 3, 4, and 5), while the amplitude of the turntable remained at 5°. This resulted in the reversal of the VOR direction, an inversion of the compensatory eye movement driven by vestibular input, moving the eye in the same direction as the head rotation instead of the normal compensatory opposite direction.

A CCD camera was fixed to the turntable in order to monitor the eyes of the mice. Eye movements were recorded with eye-tracking software (ETL-200, ISCAN systems, Burlington, NA, USA, RRID:SCR_021044). Eyes were illuminated during the experiments using two table-fixed infrared emitters (output 600 mW, dispersion angle 7°, peak wavelength 880 nm) and a third emitter, which produced the tracked corneal reflection, was mounted to the camera and aligned horizontally with the optical axis of the camera. Eye movements were calibrated by moving the camera left-right (peak-to-peak 20°) during periods that the eye did not move (*Stahl, 2004*). Gain and phase values of eye movements were calculated using custom-made MATLAB scripts, available at GitHub (GitHub, San Francisco, CA, USA, https://github.com/MSchonewille/iMove; copy archived at swh:1:rev: e0dda8be37519e58387c2b9702479625e66b54ec; *Beekhof, 2021*; *Schonewille et al., 2010*). Consolidation is the percentage of adaptive change that was still present after 23 hr in the dark and was calculated as 100%*(dxt0-dx + 1 t0)/(dxt0-dxt30), with dxt0 as the value before training on the first day, dx + 1t0 as the value before training on the next day and dxt30 as the last, final value on the first day.

## Eyeblink conditioning

Young (P21, on the first day of training) and adult (10–11 weeks old) C57BL/6J mice were used to perform EBC behavioral tests as done previously (*Boele et al., 2018*). Briefly, a metal pedestal was placed on the mice skull under general anesthesia with isoflurane/O$_2$ in order to allow for head fixation during the EBC experiments. After 3 days of recovery, mice were head-fixed and placed on top of a foam cylindrical treadmill on which they were allowed to walk freely. Mice were habituated for 2 days (30 min per day) before beginning the conditioning sessions in a sound- and light-isolating chamber which houses the eye-blink set-up. No stimuli was delivered during habituation.

Eyelid movements were monitored under infrared illumination using a high-speed (3333 frames/s) monochrome video camera. To calculate the fraction of eyelid closure (FEC), a region of interest (ROI) was selected around the eye when this was fully open (including the pupil, iris and immediate surrounding fur). The grayscale values of the pixels in the ROI were converted to binary in such way that the pupil and iris had a value of 0 and the fur a value of 1. All pixels in the ROI were summed to calculate the area of fur in all frames. In the end, the raw pixels were normalized into FEC units (from 0 – fully open eye, to 1 – fully closed eye [*Heiney et al., 2014*]). After the habituation session the EBC test lasted 5 days. Each day there were two sessions, one in the morning and one in the

afternoon spanned 6 hr. All experiments were performed at approximately the same time of day by the same experimenter. On the day of acquisition session 1, each animal first received 20 conditioned stimulus-only trials as a baseline measure, to establish that the conditioned stimulus did not elicit any reflexive eyelid closure. During each session, every animal received in total 200 paired conditioned stimulus-unconditioned stimulus trials, 20 unconditioned stimulus only trials, and 20 conditioned stimulus only trials. These trials were presented over 20 blocks, each block consisted of 1 unconditioned stimulus only trial, 10 paired conditioned stimulus-unconditioned stimulus trials, and 1 conditioned stimulus only trial. The interval between the onset of the conditioned stimulus and that of the unconditioned stimulus was set at 250 msec. The conditioned stimulus was a green LED light (conditioned stimulus duration 280 msec, LED diameter 5 mm) placed 10 cm in front of the mouse's head. The unconditioned stimulus was a weak air-puff applied to the eye (30 psi, 30 msec duration), which was controlled by a pressure injector and delivered via a needle perpendicularly positioned at 5 mm from the center of the left cornea.

Individual eyeblink traces were analyzed with a custom written script in MATLAB R2018a. Trials with significant activity in the 500 msec pre-conditioned stimulus period were regarded as invalid for further analysis. Valid trials were further normalized by aligning the 500 msec pre-conditioned stimulus baselines and calibrating the signal so that the size of a full blink was 1. In valid normalized trials, all eyelid movements larger than 0.05 and with a latency to CR onset between 50 and 250 msec, a latency to CR peak of 100–250 msec (relative to conditioned stimulus onset) and a positive slope in the 150 msec before unconditioned stimulus time were considered as conditioned responses (CRs).

## Statistical analysis

Error bars in all graphs indicate mean ± SEM. For each experiment, the sample size and statistical tests used are summarized in *Supplementary file 1*. Data was tested for normality with Shapiro–Wilk tests and for equal variances using the *F*-test. For normal distributed data, statistical significance was determined by the two-way ANOVA with multiple comparisons or mixed-effects test with repeated measures. If the data was not normally distributed, statistical significance was calculated using the Kruskal–Wallis test with multiple comparisons. The minimum level of significance accepted for all tests was $p < 0.05$. Statistical analyses were performed using GraphPad Prism (GraphPad Software, San Diego, CA, USA, RRID:SCR_002798).

## Acknowledgements

The authors kindly thank Laura Post, Sander Kruithof and Erika Sabel-Goedknegt for excellent technical assistance; Dick Jaarsma and Lynette Lim for discussions and comments on the manuscript. This work was supported by an ERC starter grant (ERC-Stg #680235; MS), Dutch Organization for Life Sciences (Off-Road fellowship; ZonMW-451001027; CO), VENI fellowship (NWO-ENW; JJW), VIDI grant (NWO-ENW; #016.121.346, FEH) and Medical Sciences (TOP-GO #91210067, FEH) and the CJ Vaillant Fund (FEH).

## Additional information

### Funding

| Funder | Grant reference number | Author |
| --- | --- | --- |
| H2020 European Research Council | ERC-Stg #680235 | Martijn Schonewille |
| ZonMw | ZonMW-451001027 | Catarina Osório |
| Nederlandse Organisatie voor Wetenschappelijk Onderzoek | #016.121.346 | Freek E Hoebeek |
| ZonMw | #91210067 | Freek E Hoebeek |
| Nederlandse Organisatie voor Wetenschappelijk Onderzoek | 016.Veni.192.270 | Joshua J White |

The funders had no role in study design, data collection and interpretation, or the decision to submit the work for publication.

## Author contributions
Gerrit Cornelis Beekhof, Conceptualization, Data curation, Formal analysis, Visualization, Methodology, Writing - original draft, Writing - review and editing; Catarina Osório, Joshua J White, Conceptualization, Data curation, Formal analysis, Funding acquisition, Visualization, Methodology, Writing - original draft, Writing - review and editing; Scott van Zoomeren, Willem Ashwin Mak, Marit Runge, Data curation, Formal analysis; Hannah van der Stok, Data curation, Formal analysis, Visualization; Bilian Xiong, Ingo HMS Nettersheim, Data curation; Francesca Romana Fiocchi, Formal analysis; Henk-Jan Boele, Formal analysis, Funding acquisition, Methodology; Freek E Hoebeek, Conceptualization, Funding acquisition, Methodology; Martijn Schonewille, Conceptualization, Formal analysis, Funding acquisition, Methodology, Writing - original draft, Writing - review and editing

## Author ORCIDs
Gerrit Cornelis Beekhof (iD) https://orcid.org/0000-0002-3038-5023
Catarina Osório (iD) https://orcid.org/0000-0002-5228-0599
Joshua J White (iD) https://orcid.org/0000-0002-6218-623X
Hannah van der Stok (iD) https://orcid.org/0000-0002-4134-4075
Ingo HMS Nettersheim (iD) http://orcid.org/0000-0001-5207-5328
Martijn Schonewille (iD) https://orcid.org/0000-0002-2675-1393

## Ethics
Animal experimentation: This study was performed under a project license approved by the Dutch Central Committee for Animal Experiments (CCD, AVD #101002015273). Each experiment was separately verified and approved by the Animal Welfare Body (IvD/AWB, various work protocols). All surgeries were performed under isoflurane anesthesia combined with local anesthetics and analgesics in an effort to minimize suffering.

## Decision letter and Author response
Decision letter https://doi.org/10.7554/eLife.63668.sa1
Author response https://doi.org/10.7554/eLife.63668.sa2

# Additional files
## Supplementary files
• Supplementary file 1. Data for *in vivo*, *in vitro* electrophysiology, anatomical and behavioral quantifications. Descriptive values are represented as mean ± SEM. p=postnatal, Z−=ZebrinII negative, Z+=ZebrinII positive, LIII = lobule III, LX = lobule X, LI-III = lobules I/II/III, LIX-X = lobules IX/X.

• Transparent reporting form

## Data availability
All data generated or analyzed during this study are included in the manuscript and supporting files.

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
