## [Decision Letter]

**Acceptance summary:**

This work is a follow-up to the authors' previous study demonstrating that the properties of Purkinje cells in zebrin+ and zebrin- compartments differ in multiple ways. Now, they have shown when these differences emerge during development, and tested whether these timelines correlate with the emergence of zebrin+ and zebrin- dependent behaviors. The experiments are technically impressive, including longitudinal evaluation of electrophysiological properties (*in vivo* and *in vitro*) and morphology of Purkinje cells, as well as analysis of cerebellum-dependent behaviors in juvenile mice directly after weaning. The results are important because they provide a first glimpse into the developmental timelines of Purkinje cells in two separate regions of the cerebellum (anterior cerebellum and nodulus), and the impact that ontogenetic differences in these timelines may have in early motor function.

**Decision letter after peer review:**

Thank you for submitting your article "Differential spatio-temporal development of Purkinje cell subsets shapes the emergence of cerebellum-dependent behaviors" for consideration by *eLife*. Your article has been reviewed by 3 peer reviewers, and the evaluation has been overseen by Ronald Calabrese as the Senior and Reviewing Editor. The following individual involved in review of your submission has agreed to reveal their identity: Fabrice Ango (Reviewer #1).

The reviewers have discussed the reviews with one another and the Reviewing Editor has drafted this decision to help you prepare a revised submission.

Summary:

This work is a follow-up to the authors' previous study demonstrating that the properties of Purkinje cells in zebrin+ and zebrin- compartments differ in multiple ways. Now, they have shown when these differences emerge during development, and tested whether these timelines correlate with the emergence of zebrin+ and zebrin- dependent behaviors. The experiments are technically impressive, including longitudinal evaluation of electrophysiologal properties (*in vivo* and *in vitro*) and morphology of Purkinje cells, as well as analysis of cerebellum-dependent behaviors in juvenile mice directly after weaning. The results are important because they provide a first glimpse into the developmental timelines of Purkinje cells in two separate regions of the cerebellum (anterior cerebellum and nodulus), and the impact that ontogenetic differences in these timelines may have in early motor function.

Essential revisions:

A number of concerns were raised by the reviewers that preclude acceptance at this stage. The paper could be substantially improved without new experiments but requiring new analysis and rewriting. The story may change quite a bit and may end up having much less to do with differences in the developmental timelines between zebrin regions, and much more to do with differences in the developmental timelines between lobules. The link between stripes compartmentalization and Purkinje cells maturation is weak since the authors choose to characterize lobules far apart in the antero-posterior axis. The paper will be no less interesting by focusing on specific lobules without mentioning their zebrin identity that can be discussed in the end. The question of when PCs reach maturity in the specific lobules is important and will clearly enhance the quality of the paper. The authors should be able to provide the exact PCs localization together with specific features in the different lobules. By reorganizing the data, they might be in position to compare lobule I, II or III together with IX and X…

The consensus of the reviewers is that the authors should address:

1. The properties of Purkinje cells in lobules involved in eyeblink conditioning reach maturity later than in lobules involved in VOR adaptation. Showing this will require new analyses since as it stands, none of the figures really show when the properties of Purkinje cells reach maturity in any lobules.

2. Eyeblink conditioning is impaired in juvenile mice at the time when the properties of Purkinje cells in the eyeblink lobules have yet to reach maturity, whereas VOR adaptation is fine at that same time. Again, showing this will also require new analyses since the current evaluation of VOR adaptation is insufficient to know whether the juvenile mice are impaired or not.

Showing 1 and 2 would provide useful information for the field, even if the results are correlational. In general, we know very little about the developmental timeline of Purkinje cell properties in different regions of the cerebellum, and the paper would speak to that. It would have been useful to look at the developmental timeline of plasticity in Purkinje cells of different lobules, but that will require new experiments and is beyond the scope of revision.

If in addition to 1 and 2, the authors were able to convincingly show that the properties of Purkinje cells in other zebrin- regions (in addition to the eyeblink regions) reach maturity late, while those in other zebrin+ regions (in addition to the VOR regions) reach maturity early, then the paper would be stronger. But showing whether their findings apply to zebrin- and zebrin+ regions, IN GENERAL, may not be possible.

Detailed comments of the reviews support these concerns and provide further guidance.

*Reviewer #1:*

In their manuscript "Differential spatio-temporal development of Purkinje cell subsets shapes the emergence of cerebellum-dependent behaviors ", Beekholf et al., combined several *in vitro* and state-of-the art *in vivo* approaches to characterize the developmental trajectories of distinct Purkinje cells (PCs) localized in Zebrin + or Zebrin – region of the cerebellum. They found that several intrinsic features of PCs development are differentially regulated in a zebrin II-dependent manner. The timing of PCs maturation in these regions (Zebrin+ or Zebrin -) is mirrored by specific cerebellum-dependent behavioral acquisition. Overall, I found the study to be very straight-forward and the presented data are fairly supportive of their conclusion, although mainly correlative.

I only have a few concerns:

1. The overall heterogeneity of PCs intrinsic activities within the same lobule can be clearly observed in Figure 1 and Figure 2. For example, some plots distribution (in Fig1C1, C2 or Supp 1B1, 1B2) appeared bimodal. A population with high SS activity (SS rate above 60hz) and one with low SS activity (SS rate Below 35hz) at P12-P17 can clearly be observed. Since the authors labelled the recording sites with biocytin, it will be interesting to relate the PCs location to PCs intrinsic activities.

2. One important statement of the manuscript is that early developmental trajectory of PCs is zebrinII-dependent. However, in early and late phase of cerebellar development, the strict identification of ZebrinII+ domain is challenging. It is not certain whether early PCs identified as Zebrin II+ are related to each of the PC ZebrinII + stripes in the adult. Although the idea is attractive and the correlative data in lobule III and X showed distinct PC features, one might argue that change in PC molecular identity during development within a specific lobule is likely. The author should discussed this possibility or add new arguments to strengthen their hypothesis.

3. PC morphology can vary depending on their location within the same lobule. One localized at the base or at the apex of the cerebellar cortex follium have significant different morphological features (A. Sudarov and A. L. Joyner 2007). The authors should indicate where the PCs were quantified and provide their overall localization within a lobule.

4. A clear identification of PC molecular signature during development is not trivial unless longitudinal clonal study can be performed in PCs. Is it possible to dismiss the existence of an anteroposterior developmental gradient of PC maturation? For example, did PCs present is lobule I showed distinct intrinsic activities from those in lobule III, both being Zebrin II+.

*Reviewer #2:*

This manuscript by Beekhof et al. examines the morphological and physiological development of zebrin + and zebrin – Purkinje cells in the cerebellum and relate this to emergence of cerebellum dependent behaviors in each region. The authors examined simple spike firing rate and regularity, complex spike rate and regularity, dendritic morphology, density of climbing fiber synapses, and axonal arborization of zebrin +/- Purkinje cells across a range of ages (P3- adult). They find that zebrin + and – Purkinje cells differ across many of these measurements (as has been observed previously), but critically, they find that zebrin – Purkinje cells typically reach mature phenotypes at younger ages compared to zebrin + Purkinje cells. From these results, the authors hypothesize that zebrin – circuits may mature more quickly. They test this by training young and old mice in VOR phase reversal (a behavior largely dependent on zebrin – circuits) and delay eyeblink conditioning (a behavior largely dependent on zebrin + circuits). Indeed, they find that young mice exhibit robust VOR learning but impaired eyeblink conditioning, consistent with earlier maturation of Zebrin – Purkinje cells.

In general, I find the data regarding the development of zebrin +/- Purkinje cells to be thorough and convincing. I am more concerned about the behavioral data and whether the behavioral differences can really be attributed to the differences observed in the first half of the paper.

My major concerns related to the use and interpretation of the behavioral experiments.

1. The authors want to draw a connection between the behavioral results and the development of zebrin +/- Purkinje cells. In my opinion, this is a tenuous connection supported by little evidence. There is a somewhat weak correlation between behavior and the measured properties of Purkinje cells, but little or nothing to suggest causation here. There are any number of other factors that could result in delayed learning of eyeblink conditioning compared to VOR adaptation. Some of these are considered in the discussion, but the link between development of zebrin +/- Purkinje cells and behavioral outcomes should be toned down.

2. Learning during VOR adaptation or eyeblink conditioning likely engages Purkinje cell plasticity (either in the form of parallel fiber LTP/LTD or changes in intrinsic excitability), yet these parameters were not examined across age or zebrin region, why not?

3. My final concern regarding the behavior experiments is with the underlying assumption that behavioral learning always improves with cellular/circuit development. The finding that VOR adaptation is more efficient in juvenile animals (compared to adults) contradicts this assumption. I don't think you can say that because VOR adaptation is faster in juveniles, that the underlying circuits have developed earlier compared to the circuits underlying eyeblink conditioning. It could be argued that the VOR circuits are in fact in an immature hyperplastic state in juvenile mice, and this is not evidence of more fully developed Purkinje cells.

*Reviewer #3:*

This work is a follow-up to the authors' previous study demonstrating that the properties of Purkinje cells in zebrin+ and zebrin- compartments differ in multiple ways. Now, they have shown when these differences emerge during development, and tested whether these timelines correlate with the emergence of zebrin+ and zebrin- dependent behaviors. The experiments are technically impressive, including longitudinal evaluation of electrophysiologal properties (*in vivo* and *in vitro*) and morphology of Purkinje cells, as well as analysis of cerebellum-dependent behaviors in juvenile mice directly after weaning. The results are important because they provide a first glimpse into the developmental timelines of Purkinje cells in two separate regions of the cerebellum (anterior cerebellum and nodulus), and the impact that ontogenetic differences in these timelines may have in early motor function.

There are a number of concerns that need to be addressed. With the exception of concern 4, however, none of these concerns require new experiments and could be adequately addressed with new analysis and text revisions.

1. One of the main claims is that the properties of Purkinje cells in zebrin+ compartments reach mature levels earlier than in zebrin- compartments. However, most of the analysis and all of the figures are focused on comparing when the differences between Purkinje cells in zebrin+ and zebrin- emerge during development. This is a different question. The analysis and the figures should be redone to assess when the properties of Purkinje cells reach maturity, rather than assessing when the properties of zebrin+ and zebrin- Purkinje cells become different from each other. It will be critical to specify the criterion used to determine whether Purkinje cell properties have reached maturity. For example, Figure 4B2 shows many more VGluT2 puncta in zebrin+ than in zebrin- Purkinje cells by age P7, but it would be wrong to assume that this means that VGluT2 expression has reached maturity in zebrin+ compartments by age P7 because the number of VGluT2 puncta undergoes extensive remodeling after that age and is dramatically reduced on P14 and P21, before starting to go up again at P35.

2. Another claim is that VOR learning is actually enhanced in juvenile mice relative to adult mice. However, the data shown is not sufficient. A more complete analysis is necessary, including: (1) a full evaluation of the eye movements of the juvenile and adult mice during the training period. Previous work has shown that the best predictor of VOR learning in mice is related to the gaze signals present during training (Shin et al., J Neurosci., 2014), and it is important to rule out differences in gaze as the root cause of the learning differences, (2) a complete evaluation of the eye movements of the juvenile and adult mice during the test periods. Currently, the analysis is restricted to assessing the phase of the learned eye movements (Figure 6A2), and there's no information about their gain, (3) To formally test whether learning is enhanced, it will be necessary to perform an analysis to determine whether the learned movements (not just the phase of those movements) are more adaptive in juvenile or adult mice. It would be interesting if it turned out that the learned movements of juvenile and adult mice achieve similar performance levels, but using different adaptive mechanisms.

3. Throughout the paper, the terms "zebrin-" and "zebrin+" are used interchangeably with the terms "anterior cerebellum" and "nodular cerebellum". In reality, however, the majority of the experiments were performed by comparing the properties of Purkinje cells in one region of the anterior cerebellum (lobules I-III) and one specific region in the nodular cerebellum (lobules IX-X). While it is true that most Purkinje cells in lobule I-III are zebrin- and most Purkinje cells in lobule IX-X are zebrin+, it unclear whether the developmental timeline of Purkinje cells in lobule I-III (or IX-X) will apply to all other zebrin+ (or zebrin-) regions of the cerebellum. For this reason, it is critical to specify the lobular location for each one of the analyses performed. It looks like all the experiments in Figure 2 were performed in only lobule III and lobule X, while experiments in Figure 3 and 4 were performed in lobules I-III and IX-X, but it is unclear whether most of the Purkinje cells were located in one of the specific lobules. In addition, there is no information about the location of the Purkinje cells in Figure 1 and Figure 5, or whether most of the Purkinje cells in these figures came from specific lobules. Unless zebrin+ and zebrin- Purkinje cells were sampled in enough numbers from all lobules, it is not possible to make general claims. For instance, if the results show that Purkinje cells in lobule III (which are zebrin-) reach maturity early on, it is incorrect to assume that zebrin- Purkinje cells in other lobules of the cerebellar cortex will also reach maturity early on. For analyses done by measuring the properties of Purkinje cells in specific lobules, the text of the paper should be revised by replacing all claims about the properties of Purkinje cells in zebrin+ and zebrin- compartments (including in the Figures) with claims about the properties of Purkinje cells in those specific lobules.

4. The regions of the cerebellar cortex involved in the two behaviors that were examined are the flocculus (for VOR adaptation) and lobules IV-V-VI (for eyeblink conditioning). The paper should report the properties and developmental timelines of Purkinje cells located in those specific lobules since it cannot be assumed that the properties and developmental timelines of Purkinje cells in lobules I-III will be the same as in IV-V-VI, or that the properties and developmental timelines of Purkinje cells in lobules IX-X will be the same as in the flocculus.

---

## [Author Response]

Reviewer #1:In their manuscript "Differential spatio-temporal development of Purkinje cell subsets shapes the emergence of cerebellum-dependent behaviors ", Beekhof et al., combined several *in vitro* and state-of-the art *in vivo* approaches to characterize the developmental trajectories of distinct Purkinje cells (PCs) localized in Zebrin+ or Zebrin– region of the cerebellum. They found that several intrinsic features of PCs development are differentially regulated in a zebrin II-dependent manner. The timing of PCs maturation in these regions (Zebrin+ or Zebrin-) is mirrored by specific cerebellum-dependent behavioral acquisition. Overall, I found the study to be very straight-forward and the presented data are fairly supportive of their conclusion, although mainly correlative.I only have a few concerns:1. The overall heterogeneity of PCs intrinsic activities within the same lobule can be clearly observed in Figure 1 and Figure 2. For example, some plots distribution (in Fig1C1, C2 or Supp 1B1, 1B2) appeared bimodal. A population with high SS activity (SS rate above 60hz) and one with low SS activity (SS rate Below 35hz) at P12-P17 can clearly be observed. Since the authors labelled the recording sites with biocytin, it will be interesting to relate the PCs location to PCs intrinsic activities.

This is indeed an interesting point. In an attempt to make these data more clear, we have plotted each recording location for P12-17 on the unfolded map of the cerebellum with a color code representing the firing rate (for both simple spikes and complex spikes) and regularity (CV and CV2). Examining these plots for patterns, we find that the location of the PC within the cerebellar cortex is not sufficient to account for differences in PC physiology. A possible exception is the firing rate in the nodulus/flocculus, which is consistently in the lower half of the simple spike range. However, in the anterior regions, and even in ZebrinII- PCs, activity levels in that lower half of the range can be found. We conclude from this analysis that, although firing frequencies appear to be bimodal, the two modes are not based on location. Only when comparing all Z+ against all Z- PCs across the cerebellar cortex can the difference in firing rate be observed. These results are now included as Figure 1 supplement 4-5 and described in the Results section (bottom, page 5).

2. One important statement of the manuscript is that early developmental trajectory of PCs is zebrinII-dependent. However, in early and late phase of cerebellar development, the strict identification of ZebrinII+ domain is challenging. It is not certain whether early PCs identified as Zebrin II+ are related to each of the PC ZebrinII + stripes in the adult. Although the idea is attractive and the correlative data in lobule III and X showed distinct PC features, one might argue that change in PC molecular identity during development within a specific lobule is likely. The author should discussed this possibility or add new arguments to strengthen their hypothesis.

We agree with the reviewer that this is an important question. Several studies provide strong evidence that together supports the concept that for most of the PCs its identity is determined at their birth and maintained into adulthood.

It should be noted that we do not claim that the early developmental trajectory of PCs is zebrinII-dependent; we used ZebrinII as a marker that labels the cerebellar-olivar compartmentalization, as traditionally described. Nevertheless, there are several papers showing that adult cerebellar compartmentalization correlates with embryonic clusters. Hashimoto and Mikoshiba (Hashimoto M, Mikoshiba K. J Neurosci. 2003) PC birthdating experiments revealed that, in contrast with the expression of the markers available at the time, the 8 clusters of PCs the authors identified embryonically were unchanged until adulthood. Although ZebrinII is a “late-onset” parasagittal marker, others, such as neurogranin (Larouche M, Che P, Hawkes R. J Comp Neurol. 2006), were shown to be expressed from E14.5 in a pattern that was maintained until P20. While EAAT4 expression is similar to that of ZebrinII in adult mice, its mRNA can already by detected from embryonic ages and immunohistochemical analysis supports a patterned, selective expression in the caudal cerebellum from embryonic day 18 (Yamada K, Wada S, et al. Neurosci Res. 1997). Additionally, PLCβ4 expression is restricted to the Z- cells and its expression also begins just before birth. The PLCβ4 positive clusters in the neonate are complementary with Z- PCs (Marzban H, Chung S, Watanabe M, Hawkes RJ. Comp Neurol. 2007). With the use of mouse genetics (Sillitoe RV, Gopal N, Joyner AL. Neuroscience. 2009) or adenovirus tracing (Namba K, Sugihara I, Hashimoto M. J Comp Neurol. 2011) it was possible to verify that the birth date-related PCs correlated with the zonal pattern of ZebrinII. Finally, it is known that the nodulus (vermal lobule X) and the flocculus are distinct divisions of the cerebellum at an early age (Hirofumi Fujita, Noriyuki Morita, Teiichi Furuichi and Izumi Sugihara. The Journal of Neuroscience, 2012).

Taken together, the existing evidence supports the concept most PCs obtain its identity (or ‘fate’) at their birth, or shortly after. This more extended evaluation of the issue is now included in the discussion (page 14).

3. PC morphology can vary depending on their location within the same lobule. One localized at the base or at the apex of the cerebellar cortex follium have significant different morphological features (A. Sudarov and A. L. Joyner 2007). The authors should indicate where the PCs were quantified and provide their overall localization within a lobule.

Indeed, Sudarov and Joyner, 2007, and also e.g. Nedelescu and colleagues (2013, 2018), demonstrated that Purkinje cell morphology correlates with its location with respect to the fissures and shape of the lobules. Our study focuses on the difference between cerebellar modules, which run across lobules and thus include Purkinje cells in the apex, base and sulcus. However, as indicated by the newly included figure 3 supplement 1 B1-2, there was no systematic bias in our sampling from apex, base and sulcus, arguing against the possibility that a selection bias explains the differences observed in Figure 3.

We also did not test for e.g. the number of primary dendrites, as they did, but rather focused on the number of intersections and size markers, like max. dendrite length and area. In line with Nedelescu et al., 2018, we find that from P12 to P60 the change in area for LIX-X is small, while in more anterior regions (LV in that paper, Figure 5 “mean hull”; LIII in ours) the increase is larger, resulting in a significant difference in adult PCs compared to LIX-X.

We have now included the location within the lobule for all cells analyzed, separated by apex, sulcus or base, as Figure 3 supplement1 A-B. In addition, we have included graphs to indicate the lobule of origin for all cells included in this analysis and address these figures in the Results section (page 7).

4. A clear identification of PC molecular signature during development is not trivial unless longitudinal clonal study can be performed in PCs. Is it possible to dismiss the existence of an anteroposterior developmental gradient of PC maturation? For example, did PCs present is lobule I showed distinct intrinsic activities from those in lobule III, both being Zebrin II+.

This is an interesting suggestion. We considered the possibility of lobule-related differences in our previous work (Zhou et al., 2014) and made an effort to answer it by performing a comprehensive analysis of firing rates over all lobules (see Figure 3 of that paper). This required the recording of PC activity in 245 cells in adult mice with individual labeling and postmortem analysis, and we found no evidence for lobule-specific differences. Instead, we observed a correlation between the averaged zebrin intensity across all PCs and the averaged firing rate of the recorded PCs, per lobule. Performing such an analysis throughout mouse development by recording a similar number of cells at different ages, would multiply this number several times. However, to address this concern, we have now compared the results across lobules for each group (Figure 1 supplement 3 A1-4) and made graphical representations of the simple spikes (SS) and complex spikes (CS) firing rates in color code, for P12 to P17, on the unfolded cerebellar map (Figure 1 supplement 4). We find no evidence for a clear pattern in the development of activity in either the anteroposterior, or the mediolateral axis.

This point is now addressed in detail in the Results section (middle page 5), based on the new figure 1 supplements3-5.

Reviewer #2:This manuscript by Beekhof et al. examines the morphological and physiological development of zebrin + and zebrin – Purkinje cells in the cerebellum and relate this to emergence of cerebellum dependent behaviors in each region. The authors examined simple spike firing rate and regularity, complex spike rate and regularity, dendritic morphology, density of climbing fiber synapses, and axonal arborization of zebrin +/- Purkinje cells across a range of ages (P3- adult). They find that zebrin + and – Purkinje cells differ across many of these measurements (as has been observed previously), but critically, they find that zebrin – Purkinje cells typically reach mature phenotypes at younger ages compared to zebrin + Purkinje cells. From these results, the authors hypothesize that zebrin – circuits may mature more quickly. They test this by training young and old mice in VOR phase reversal (a behavior largely dependent on zebrin – circuits) and delay eyeblink conditioning (a behavior largely dependent on zebrin + circuits). Indeed, they find that young mice exhibit robust VOR learning but impaired eyeblink conditioning, consistent with earlier maturation of Zebrin – Purkinje cells.In general, I find the data regarding the development of zebrin +/- Purkinje cells to be thorough and convincing. I am more concerned about the behavioral data and whether the behavioral differences can really be attributed to the differences observed in the first half of the paper.My major concerns related to the use and interpretation of the behavioral experiments.1. The authors want to draw a connection between the behavioral results and the development of zebrin +/- Purkinje cells. In my opinion, this is a tenuous connection supported by little evidence. There is a somewhat weak correlation between behavior and the measured properties of Purkinje cells, but little or nothing to suggest causation here. There are any number of other factors that could result in delayed learning of eyeblink conditioning compared to VOR adaptation. Some of these are considered in the discussion, but the link between development of zebrin +/- Purkinje cells and behavioral outcomes should be toned down.

We understand the concern of the reviewer. We attempted to present the differences in the development of cellular features as a motivation to study the related behaviors, but agree that our experiments do not prove causality. We have made two adjustments based on these comments, and those of reviewer 3:

1. We included a new analysis, as Figure 1 supplement 3B, that depicts the development of the SS activity of Z- PCs in the anterior vs. the hemispheral regions and that of Z+ PCs in the flocculus vs. the nodulus. When we compare the PC SS activity at the ages that the behavioral experiments were done (young: P18-P29 vs. adult: >P60), we find a significant increase in the Z- PCs of the hemispheres, and no suggestion for a change in Z+ floccular PCs, although the numbers are low.

2. We toned down the wording where we discuss the rationale for the experiment and the interpretation of the outcome.

These clarifications include a change of the title, change in the last part of the abstract, change in the last paragraph of the introduction and the related sections in the results (page 9) and the discussion (page 15).

2. Learning during VOR adaptation or eyeblink conditioning likely engages Purkinje cell plasticity (either in the form of parallel fiber LTP/LTD or changes in intrinsic excitability), yet these parameters were not examined across age or zebrin region, why not?

We agree with the reviewer that it would be very interesting to determine the developmental timeline of various forms of plasticity and if and how they contribute to both types of cerebellum-dependent learning. Based on current literature it is very likely that there are indeed regional or ZebrinII-related differences. For instance, Wadiche and Jahr (2005) showed that LTD could readily be induced in lobule III, but not in lobule X. A manuscript archived last year indicates that the potential for the induction of LTP and the plasticity of intrinsic excitability are both stronger Z- PCs, in P16-24 mice (Nguyen-Minh et al., 2020). However, recent experiments by Suvrathan et al., (2018) indicate that in the flocculus the timing of the climbing fiber stimulation relative to that of the parallel fiber is crucial for LTD induction and that the optimal timing for LTD varies across regions, indicating that, at least for LTD but perhaps also for the other forms, induction of plasticity requires the precise selection of parameter, and these parameters differ across regions.

Taken together, we believe that a systematic study of LTD, LTP and plasticity of intrinsic excitability, with potential region- or zebrinII-related key parameters and across different ages, would constitute a separate manuscript paper by itself. However, we do agree that this is an important issue to address and have included this consideration of the potential role of differentiation of plasticity mechanisms in the new manuscript (discussion page 16)

3. My final concern regarding the behavior experiments is with the underlying assumption that behavioral learning always improves with cellular/circuit development. The finding that VOR adaptation is more efficient in juvenile animals (compared to adults) contradicts this assumption. I don't think you can say that because VOR adaptation is faster in juveniles, that the underlying circuits have developed earlier compared to the circuits underlying eyeblink conditioning. It could be argued that the VOR circuits are in fact in an immature hyperplastic state in juvenile mice, and this is not evidence of more fully developed Purkinje cells.

We agree with the reviewer that the hyperplastic state of VOR adaptation could be considered a sign that the circuit has not reached its stable, mature state yet. In that case, the term ‘(im)mature’ perhaps does not convey the meaning we had intended with respect to the behavioral changes (we believe it is suitable for other aspects of PC development). The objective of using the term was to make clear that Z+ PCs reached a state with more optimal performance at the juvenile stage, in contrast to the circuit underlying eyeblink conditioning which at that point functioning sub-optimally. It would be interesting to determine the complete developmental timeline of eyeblink conditioning. It is tempting to speculate, but based on the existing literature an earlier peak of hyperplasticity in eyeblink conditioning seems improbable (Freeman, 2014).

To address this concern, we have now reworded our description of the relationship between the behavioral data and the development of PC features and included the concept of an immature hyperplasticity in the discussion (both page 16).

Reviewer #3:This work is a follow-up to the authors' previous study demonstrating that the properties of Purkinje cells in zebrin+ and zebrin- compartments differ in multiple ways. Now, they have shown when these differences emerge during development, and tested whether these timelines correlate with the emergence of zebrin+ and zebrin- dependent behaviors. The experiments are technically impressive, including longitudinal evaluation of electrophysiologal properties (*in vivo* and *in vitro*) and morphology of Purkinje cells, as well as analysis of cerebellum-dependent behaviors in juvenile mice directly after weaning. The results are important because they provide a first glimpse into the developmental timelines of Purkinje cells in two separate regions of the cerebellum (anterior cerebellum and nodulus), and the impact that ontogenetic differences in these timelines may have in early motor function.There are a number of concerns that need to be addressed. With the exception of concern 4, however, none of these concerns require new experiments and could be adequately addressed with new analysis and text revisions.1. One of the main claims is that the properties of Purkinje cells in zebrin+ compartments reach mature levels earlier than in zebrin- compartments. However, most of the analysis and all of the figures are focused on comparing when the differences between Purkinje cells in zebrin+ and zebrin- emerge during development. This is a different question. The analysis and the figures should be redone to assess when the properties of Purkinje cells reach maturity, rather than assessing when the properties of zebrin+ and zebrin- Purkinje cells become different from each other. It will be critical to specify the criterion used to determine whether Purkinje cell properties have reached maturity. For example, Figure 4B2 shows many more VGluT2 puncta in zebrin+ than in zebrin- Purkinje cells by age P7, but it would be wrong to assume that this means that VGluT2 expression has reached maturity in zebrin+ compartments by age P7 because the number of VGluT2 puncta undergoes extensive remodeling after that age and is dramatically reduced on P14 and P21, before starting to go up again at P35.

We recognize that we insufficiently supported our claims regarding the temporal changes within the specific subpopulations. We want to stress that the primary goal of the experiments was to analyze when two largely overlapping representations of the same neural subpopulations started to differ in their neuronal features such as activity and morphology. In analyzing our results, we realized that particular features such as simple spike activity and dendritic complexity reached adult levels earlier in the one, compared to the other subpopulation. For example, we now show that the *in vivo* simple spike firing rate of Z- PCs in all age groups (P12-P17, P18-P29 and P30-59) is significantly lower than that in adults (>P60, all p < 0.01). In contrast, for Z+ PCs only the simple spike rate of P12-P17 PCs is lower (p < 0.0001), the other two age groups are not significantly different from the adult group (both p > 0.15). We now included these statistics whenever a comparison of the temporal development is made, i.e. for simple spike and complex spike rate and dendritic intersections. The fact that both the number of samples as well as the variance for each group is comparable between Z+ and Z- recordings supports this approach.

Thus, considering these important reviewer comments and the experimental design, we opted to consistently first present the data of the comparative analyses between different subpopulations. Then, for a selection of parameters, we provide a careful description of the changes within subpopulations over time with the statistical analyses in the respective parts of the Results sections. All new statistical analyses are also included in Suppl. Table 1.

2. Another claim is that VOR learning is actually enhanced in juvenile mice relative to adult mice. However, the data shown is not sufficient. A more complete analysis is necessary, including: (1) a full evaluation of the eye movements of the juvenile and adult mice during the training period. Previous work has shown that the best predictor of VOR learning in mice is related to the gaze signals present during training (Shin et al., J Neurosci., 2014), and it is important to rule out differences in gaze as the root cause of the learning differences, (2) a complete evaluation of the eye movements of the juvenile and adult mice during the test periods. Currently, the analysis is restricted to assessing the phase of the learned eye movements (Figure 6A2), and there's no information about their gain, (3) To formally test whether learning is enhanced, it will be necessary to perform an analysis to determine whether the learned movements (not just the phase of those movements) are more adaptive in juvenile or adult mice. It would be interesting if it turned out that the learned movements of juvenile and adult mice achieve similar performance levels, but using different adaptive mechanisms.

We appreciate this suggestion for more details on the VOR adaptation results, which we think are very interesting. The comments are addressed point-by-point:

1. We have now included the analysis of gaze during the training (Figure 6 – supplement 2D-E). It should be noted that in purely visual OKR, there is no difference between juvenile and adult mice (Figure 6 – suppl.1). In contrast, the VOR at baseline, before training, is lower in juvenile mice (0.48 vs. 0.71 in adults). However, already in the first 5 minutes of training the difference in gaze (represented by the gain relative to table, panel D) is a lot smaller, 0.18 vs. 0.23, in juvenile vs. adult. Moreover, from day 2 onwards the visual stimulus aims to reverse the VOR. Juvenile mice have a better performance in tracking the visual stimulus during learning, but the differences between juvenile and adult are smaller than in the VOR ‘probe’ trials with only vestibular stimulus (e.g. max. difference in gaze phase during training <40°, max. difference in probe VOR >100°). In an additional analysis, we determined the consolidation of adapted responses overnight, a parameter that should not depend on gaze. We found a robust difference in both gain and phase consolidation, with better consolidation in juvenile mice (included in Figure 6 supplement 2D-E). Thus, a contribution of gaze differences cannot be excluded, but the difference in consolidation appears to be a robust factor.

2. We have now also included the gain data, separately in Figure 6 supplement 2B, and combined with phase in a polar plot in Figure 6 supplement 3A-B. The gain also differs between juvenile and adult mice, particularly in the first three days. On days 4 and 5, the gain is no longer different, but the difference in phase indicates a persistence of the difference, also visible in the polar plot.

3. We now provide a more complete analysis of the adaptive response in juvenile and adult mice. In addition to the new panels described above, we included example traces. No consistent differences in e.g. the shape of the adaptive eye movements were observed, apart from the differences in gain and phase. We did observe a difference in the ‘route’ or ‘tactics’ employed by juvenile and adult mice, visualized by the polar plots. Panel A depicts gain/phase values per mouse per day and reveals that juvenile mice decrease their VOR gains more, before reversing the phase, visualized by crossing the y-axis (90 degr) at lower gain values.

To summarize, we have included new figures (Figure 6 supplement 2 and 3) and discussed these results in the main text (page 10-11).

3. Throughout the paper, the terms "zebrin-" and "zebrin+" are used interchangeably with the terms "anterior cerebellum" and "nodular cerebellum". In reality, however, the majority of the experiments were performed by comparing the properties of Purkinje cells in one region of the anterior cerebellum (lobules I-III) and one specific region in the nodular cerebellum (lobules IX-X). While it is true that most Purkinje cells in lobule I-III are zebrin- and most Purkinje cells in lobule IX-X are zebrin+, it unclear whether the developmental timeline of Purkinje cells in lobule I-III (or IX-X) will apply to all other zebrin+ (or zebrin-) regions of the cerebellum. For this reason, it is critical to specify the lobular location for each one of the analyses performed.

We understand this point raised by the reviewer. Throughout the manuscript, we attempted to only link *in vivo* activity and axonal development directly to ZebrinII-identity, while *in vitro* activity, morphology and CF inputs were discussed in terms of selected lobules. However, we understand that the use of both gives the sense that the two are interchangeable and that e.g. PC activity *in vitro* in lobule III represents all Z- PCs. For dendritic morphology and CF input, we agree it is not known whether results from specified lobules represent all Z+ or Z- PCs. However, considering the (electro)physiological data, it should be noted that previous work presents several arguments that support using lobules III and X as proxies for Z- and Z+ PCs, respectively. For instance, when Z+ and Z- PC firing rates are compared for each transverse zone, anterior, central, posterior and vestibular, the same difference was observed as that was found between lob. III and X (Zhou et al., 2014, Figure 2E). Moreover, comparison of Z+ and Z- PC firing rates in lobules VIII/IXa, Crus IIa and the C1 and A2 zones (Xiao et al., 2014), as well as targeted 2p-imaging based recordings from Z+ and Z- PCs in lobules V/VI (Zhou et al., 2014; Wu et al., 2019), also consistently demonstrated similar differences to those described for lob. III vs. lob. X. Moreover, the difference between lobules III and X, in the absence of input using slice recordings, was confirmed in PCs recorded randomly throughout the cerebellar cortex and categorized by EAAT4 expression. Due to the additional factor of developmental age we included in this study, repeating those control experiments was not an option.

Hence, we opted to provide more insight into the regional developmental profile by including additional figures for the simple spike firing rate per lobule per age group (Figure 1 – supplement 3A). In addition, we also focused on the P12-P17 age group by plotting all the recordings at this age on the unfolded cerebellar map with a color-coded firing rate (Figure 1 – supplement 4). Although our sample size is insufficient for strong statements on these data, a few important features can be identified. First, at P12-P17, PCs in lobule X have a lower firing rate than those in the other lobules. This could be considered evidence that lobule X activity levels are the exclusive cause for the low rate, but figure 1 – supplement 3 indicates that there are also PCs with very low firing rates in the hemispheres (not included in supplement 2 due to low numbers). Second, the difference between Z+ and Z- become progressively clearer from P18 to adulthood. The differences between the anterior and nodular lobules is significant from P30, but the size of the difference increases towards adulthood. The cause for the slow change is the increase in the firing rate of Z- PCs. These results are included as Figure 1 – supplement 2 and described in the Results section (page 5) and discussion (page 12).

Regarding dendritic morphology and CF input, it is indeed unclear whether our results represent all Z+ or Z- PCs. We have included supplementary figures to detail the exact locations of the recordings, and make sure to discuss these results accurately (see also below, Figure 3 – supplement 1 and start of respective paragraphs).

It looks like all the experiments in Figure 2 were performed in only lobule III and lobule X, while experiments in Figure 3 and 4 were performed in lobules I-III and IX-X, but it is unclear whether most of the Purkinje cells were located in one of the specific lobules.

The lobular location of the PC data reported in Figures 3 is now indicated in Figure 3 – supplement 1C. In short, nearly all PCs in the group I-III are taken from lobule III, nearly all in the group IX-X are from lobule X. Regarding Figure 4, for each cerebellar section, images were systematically taken from lob I, II, III, IX and X.

In addition, there is no information about the location of the Purkinje cells in Figure 1 and Figure 5, or whether most of the Purkinje cells in these figures came from specific lobules.

This information is now included for Figure 1 in supplement 3 and for P12-P19 in more detail in supplement 4. For Figure 5, well-isolated single axons were examined within the cerebellar nuclei of thick sections. PCs were randomly distributed across the cerebellar cortex and not within any specific lobules. The Z+/Z- identity of the axon was classified based on the location within the cerebellar nuclei and zebrinII staining (see also Sugihara, 2011). Axons from all nuclei, medial, interposed, dentate and vestibular were included in the final sample. In addition to increasing the number of analyzed axons in Figure 5, we have now also included example images in Figure 5 supplement 1 to provide more clarity into the distribution of labeled PCs.

Unless zebrin+ and zebrin- Purkinje cells were sampled in enough numbers from all lobules, it is not possible to make general claims. For instance, if the results show that Purkinje cells in lobule III (which are zebrin-) reach maturity early on, it is incorrect to assume that zebrin- Purkinje cells in other lobules of the cerebellar cortex will also reach maturity early on. For analyses done by measuring the properties of Purkinje cells in specific lobules, the text of the paper should be revised by replacing all claims about the properties of Purkinje cells in zebrin+ and zebrin- compartments (including in the Figures) with claims about the properties of Purkinje cells in those specific lobules.

We understand the reviewer’s concern. The text has been edited to clarify the specific lobules examined. Ideally, we would have examined features such as dendritic morphology throughout all lobules and zebrinII subpopulations, but we have chosen the selected lobules as representative samples. There is substantial evidence supporting the use of lobules III and X to study differences in SS firing rate (see explanation above), as this difference was indeed confirmed to be directly coupled to zebrinII identity, not to the lobule. For *in vivo*electrophysiological recordings and the development of axonal arbors, examined PCs are randomly sampled and directly coupled to zebrinII identity. However, with respect to the development of the dendritic morphology and CF innervation, this was not possible in our experimental design.

To address this concern, we restricted the use of the term Z+ and Z- to those experiments where the identity was directly confirmed. In discussing the results, we have used the appropriate terms (e.g. lobule III, nodular, etc.) to be as clear as possible on the origin of the observed results. These changes have been made in the Results sections, including the paragraph header (page 7-8), in Figure 7 and throughout the discussion (e.g. page 13)

4. The regions of the cerebellar cortex involved in the two behaviors that were examined are the flocculus (for VOR adaptation) and lobules IV-V-VI (for eyeblink conditioning). The paper should report the properties and developmental timelines of Purkinje cells located in those specific lobules since it cannot be assumed that the properties and developmental timelines of Purkinje cells in lobules I-III will be the same as in IV-V-VI, or that the properties and developmental timelines of Purkinje cells in lobules IX-X will be the same as in the flocculus.

We agree that ideally, we would have used two behavioral paradigms that are directly and exclusively dependent on lobules III and X. Unfortunately, we are not aware of any behavioral paradigms that meet those criteria. Hence, we used two of the very few cerebellum-dependent behaviors that can be linked to either one of the two populations and have been demonstrated to be sensitive to subpopulation-specific cellular phenotypes. Retracing our steps and recording the developmental timeline of *in vivo* / *in vitro* PC firing rate and dendritic and axonal morphology specifically in these two functional areas is not feasible, but also because particularly the region related to eyeblink conditioning is not very clearly delineated in the cortex.

However, our current dataset does allow us to selectively look at subpopulations in the direct vicinity of the eyeblink and eye movement regions, and evaluate if the timelines match those of the complete Z- and Z+ population, respectively. For this analysis, we resorted to the parameter putatively most directly linked with behavioral output, the PC simple spike firing rate recorded *in vivo*. This analysis is now included as Figure 1 supplement 3B. In short, we find that when comparing the timeline of simple spike firing rate development in the hemispheres with that for lobule I-V, we find a similar pattern, with a significant and large increase in rate from P18-29 to adult. In contrast, when we compare the developmental timeline of PC activity *in vivo* in the flocculus and that of lobule X, we see no change from P18-29 to adult, in both groups.

This analysis is now included as Figure 1 supplement 3B and addressed in the Results section (page 9) and discussion (page 15-16).